# A structurally conserved site in AUP1 binds the E2 enzyme UBE2G2 and is essential for ER-associated degradation

Christopher E. Smith[1�], Yien Che Tsai[1�]*, Yu-He Liang[2¤], Domarin Khago[2], Jennifer Mariano[1], Jess Li[2], Sergey G. Tarasov[2], Emma Gergel[1], Borong Tsai[1], Matthew Villaneuva[1], Michelle E. Clapp[3], Valentin Magidson[4], Raj Chari[3], R. Andrew Byrd[2], Xinhua Ji[2], Allan M. Weissman[1]*

1 Laboratory of Protein Dynamics and Signaling, Center for Cancer Research, NCI, National Institutes of Health, Frederick, Maryland, United States of America, 2 Center for Structural Biology, Center for Cancer Research, NCI, National Institutes of Health, Frederick, Maryland, United States of America, 3 Genome Modification Core, Frederick National Laboratory for Cancer Research, Frederick, Maryland, United States of America, 4 Optical Microscopy and Analysis Laboratory, Frederick National Laboratory for Cancer Research, Frederick, Maryland, United States of America

☙ These authors contributed equally to this work.
¤ Current Address: RCSB Protein Data Bank, Rutgers, The State University of New Jersey, Piscataway, New Jersey, United States of America
* tsaiyien@mail.nih.gov (YCT); weissmaa@mail.nih.gov (AMW)

**Data Availability Statement:** Crystallography coordinates have been deposited with the RCSB protein data bank (PDB entry 7LEW). NMR spectra used to build Fig 5A–5C and a table of chemical

## Abstract

Endoplasmic reticulum–associated degradation (ERAD) is a protein quality control pathway of fundamental importance to cellular homeostasis. Although multiple ERAD pathways exist for targeting topologically distinct substrates, all pathways require substrate ubiquitination. Here, we characterize a key role for the UBE2G2 Binding Region (G2BR) of the ERAD accessory protein ancient ubiquitous protein 1 (AUP1) in ERAD pathways. This 27-amino acid (aa) region of AUP1 binds with high specificity and low nanomolar affinity to the backside of the ERAD ubiquitin-conjugating enzyme (E2) UBE2G2. The structure of the AUP1 G2BR (G2BR$^{AUP1}$) in complex with UBE2G2 reveals an interface that includes a network of salt bridges, hydrogen bonds, and hydrophobic interactions essential for AUP1 function in cells. The G2BR$^{AUP1}$ shares significant structural conservation with the G2BR found in the E3 ubiquitin ligase gp78 and in vitro can similarly allosterically activate ubiquitination in conjunction with ERAD E3s. In cells, AUP1 is uniquely required to maintain normal levels of UBE2G2; this is due to G2BR$^{AUP1}$ binding to the E2 and preventing its rapid degradation. In addition, the G2BR$^{AUP1}$ is required for both ER membrane recruitment of UBE2G2 and for its activation at the ER membrane. Thus, by binding to the backside of a critical ERAD E2, G2BR$^{AUP1}$ plays multiple critical roles in ERAD.

## Introduction

The ubiquitin and proteasome-dependent degradation of proteins from the endoplasmic reticulum (ER) via endoplasmic reticulum–associated degradation (ERAD) is a critical mechanism

shifts used to monitor and analyze the titration in these spectra (Fig 5D and 5E) have been deposited in the Biological Magnetic Resonance Data Bank (https://doi.org/10.13018/bmrbig33). Points plotted in titrations (Fig 5D and 5E) are in S1 Table. Raw data for qPCR experiments (Figs S1C, S2D, and Fig 7B) are in S3, S4 and S8 Datas. Raw data for ITC, MST and FP (S3A–S3C Fig) are in S5, S6 and S7 Datas. Raw data for protein gel electrophoresis experiments in Figs 1–8 are in S1 Data. Raw data for protein gel electrophoresis experiments in S1–S6 Figs are in S2 Data.

**Funding:** This Research was supported in whole by the Center for Cancer Research, National Cancer Institute, National Institutes of Health Intramural Research Program project numbers ZIA BC010292 (Allan M. Weissman), ZIA BC010326 (Xinhua Ji), and ZIA BC011131 (R. Andrew Byrd) and by federal funds from the National Cancer Institute, National Institutes of Health, under contract HHSN261200800001E (Leidos Biomedical Research). The funders had no role in study design, data collection and analysis, or preparation of the manuscript.

**Competing interests:** The authors have declared that no competing interests exist.

**Abbreviations:** α1AT, alpha1-antitrypsin; aa, amino acid; AUP1, ancient ubiquitous protein 1; CHX, cycloheximide; CSP, chemical shift perturbation; ER, endoplasmic reticulum; ERAD, endoplasmic reticulum–associated degradation; FP, fluorescence polarization; G2BR, UBE2G2 Binding Region; GS, Glutathione Sepharose; HMG-CoA, 3-hydroxy-3-methylglutaryl CoA; HMGCR, 3-hydroxy-3-methylglutaryl CoA reductase; ITC, isothermal titration calorimetry; KO, knockout; LPDS, lipoprotein-deficient serum; MCS, multiple cloning site; MST, microscale thermophoresis; NHK, Null Hong Kong; NMR, nuclear magnetic resonance; PCR, polymerase chain reaction; qPCR, quantitative polymerase chain reaction; RMSD, root–mean–square deviation; RNAi, RNA interference; SIM, structured illumination microscopy; SSRL, Stanford Synchrotron Radiation Lightsource; U7BR, Ubc7-Binding Region; WT, wild type.

for both protein quality control and regulation of protein levels. Dysfunction of this homeostatic mechanism is associated with a wide range of pathologies. ERAD involves the tightly coupled processes of protein recognition, ubiquitination, retrotranslocation or dislocation, and proteasomal degradation. Central to this process are specific pairs of ubiquitin protein ligases (E3s) and ubiquitin-conjugating enzymes (E2s) [1–4].

In the yeast *Saccharomyces cerevisiae*, where this process has been most extensively studied, there are two primary ERAD E3s, Hrd1p and Doa10p, both of which are polytopic RING-type E3s that are resident to the ER [5–7]. These two E3s are central to ERAD complexes that recognize substrates depending on the topology of their degradation-targeting signals (degrons). Hrd1p is generally responsible for targeting proteins for ERAD that have luminal or ER membrane degrons (ERAD-L and ERAD-M, respectively), while Doa10p has been implicated primarily in the targeting of proteins with degrons in their cytosolic domains (ERAD-C) [8–11]. Hrd1p and Doa10p function in ERAD primarily with two E2s: Ubc6p, which is carboxyl-terminally ER membrane anchored, and Ubc7p, which is not membrane bound [12,13]. Ubc7p associates with the ER membrane via interactions with the ERAD accessory protein Cue1p [14], which is anchored to the ER by a single N-terminal transmembrane domain. Cue1p also contains a cytoplasmic ubiquitin-binding CUE domain that plays a role in degradation of some substrates and has been shown to facilitate ubiquitin chain elongation [15,16]. The carboxyl-terminal Ubc7-Binding Region (U7BR) of Cue1p binds to the "backside" of Ubc7p—an area that is distinct from both the RING domain–interacting region of the E2 and its catalytic Cys and surrounding residues [17–19]. Importantly, the U7BR can activate ubiquitination in vitro by allosterically increasing the affinity of Ubc7p for the RING domains of both Hrd1p and Doa10p [19].

In mammals, there may be up to two dozen E3s that are resident to the ER and potentially involved in ERAD. Approximately half of these have been implicated in the degradation of either naturally occurring or model substrates [1,20–22]. We and others have characterized the requirements for the function of the E3 gp78 (aka AMFR or RNF45) in ERAD [17,23–28]. This polytopic ER protein has an extended carboxyl-terminal cytosolic region that includes three domains critical for its activity [23]. These include a RING domain, a CUE domain that robustly binds ubiquitin, and a binding site for the E2 UBE2G2, which is the mammalian ortholog of Ubc7p [29]. This E2 binding site in gp78 is referred to as the UBE2G2 Binding Region (G2BR) [23,30]. The gp78 G2BR (G2BR$^{gp78}$), analogous to the U7BR of yeast Cue1p, binds to the backside of UBE2G2, increases E2:RING affinity, and stimulates ubiquitination in vitro [17]. However, the functions that it performs in cells in facilitating ERAD have not been directly assessed. Unlike gp78, neither the ortholog of yeast Hrd1p, HRD1/Synoviolin [31–33], nor that of Doa10p, MARCH6/TEB4 [32,34], encode a G2BR-like region or CUE domain. Similarly, such domains have not been described for other ERAD E3s. The question then arises as to whether there is a mammalian equivalent of yeast Cue1p.

A candidate to play a Cue1p-like role in mammals is ancient ubiquitous protein 1 (AUP1). This 410-amino acid (aa) protein is inserted into both the ER membrane and lipid droplets through an N-terminal hydrophobic "hairpin" sequence. The cytoplasmic region of AUP1 includes both an acyltransferase and a CUE domain, which is followed by a carboxyl-terminal G2BR (G2BR$^{AUP1}$) [35–38]. AUP1 has been suggested to play a role in the retrotranslocation step for ERAD-L substrates of HRD1, including the Null Hong Kong (NHK) variant of alpha1-antitrypsin (α1AT) and a truncation mutant of Ribophorin I (RI$_{332}$) [35]. Although an RNA interference (RNAi) screen did not recapitulate a requirement for AUP1 in HRD1-mediated ERAD-L [39], a more recent CRISPR/Cas-9 screen did implicate AUP1 in this pathway [40]. This CRISPR/Cas-9 screen, as well as a fluorescence insertional mutagenesis screen [41], also found AUP1 to be required for degradation of fluorescent cytosolic proteins that can

associate with the ER membrane. These fluorescent substrates engage the ERAD-C machinery as a consequence of being fused to the well-described hydrophobic CL1 degron [42–46]. In both of these screens, TRC8/RNF139 was found to be an E3 for these ERAD-C substrates [40,41] and, in the insertional mutagenesis screen, another E3 MARCH6/TEB4 was also implicated [41]. In contrast to findings for ERAD-L and ERAD-C substrates, AUP1 is not required for degradation of INSIG-1, an ERAD-M substrate targeted by gp78 [40,47].

Although it is evident that AUP1 plays a role in ERAD, how it functions and whether it is involved in substrate ubiquitination have not been assessed. Lipid droplets have been postulated as an alternative to proteinaceous channels as a mechanism for dislocating proteins from the ER during ERAD [48], and mutations in the AUP1 acyltransferase domain reduce lipid droplet formation in cells loaded with oleic acid [35]. However, at least in yeast, deficiencies in lipid droplet formation do not correlate with defective ERAD [49]. Until now, the significance of the AUP1 acyltransferase domain in ERAD has not been evaluated. There is evidence that the AUP1 CUE domain provides a means of interaction with components in the HRD1 ERAD pathway as well as with misfolded proteins [35]. This interaction likely occurs by binding ubiquitinated proteins. The CUE domain has also been shown to play a role in lipid droplet clustering [36] and in ubiquitination of AUP1 itself [35]. Again, however, there has not been a direct assessment of the significance of this domain in ERAD. Along the same lines, while AUP1 is found to associate with UBE2G2 in a G2BR$^{AUP1}$-dependent manner [35,37,50], it has not been determined whether G2BR$^{AUP1}$ or its interaction with UBE2G2 is of significance in ERAD.

We now provide evidence demonstrating that AUP1 is required for UBE2G2-mediated substrate ubiquitination. By contrast, both AUP1 and UBE2G2 are dispensable for the constitutive ubiquitination of HRD1 that we observe in cells. Furthermore, while ERAD of both HRD1 (ERAD-L) and TRC8 (ERAD-C) substrates are seemingly unaffected by the loss of the acyltransferase and CUE domains, the G2BR is absolutely required for degradation of these substrates. We have similarly found requirements for G2BR$^{AUP1}$ and for G2BR$^{gp78}$ in the degradation of ERAD-M substrates for which HRD1 and gp78, respectively, are implicated. We also report the structure of the G2BR$^{AUP1}$ in complex with UBE2G2, confirm intermolecular contacts critical for this high-affinity interaction, and assess the role of G2BR$^{AUP1}$ in UBE2G2:RING affinity. Additionally, we have determined that the G2BR$^{AUP1}$ facilitates the critical role of UBE2G2 in ERAD through multiple mechanisms including increasing its levels, recruiting it to the ER membrane, and markedly enhancing its ubiquitin-conjugating activity.

## Results

### G2BR$^{AUP1}$ is required for ERAD

To assess the requirement of AUP1 in HRD1-dependent ERAD, we examined the degradation of a known ERAD-L HRD1 substrate, the NHK variant of α1AT, in HT1080 fibrosarcoma cells. The NHK variant of α1AT (hereafter referred to as NHK) contains a frameshift mutation that generates a nonsense codon, resulting in the translation of a truncated protein that is retained in the ER lumen and targeted for ERAD [35,51–53]. Consistent with the HRD1 pathway regulating NHK degradation [51], CRISPR/Cas-9 knockout (KO) of HRD1 in HT1080 cells (HRD1 KO) inhibited the degradation of NHK as assessed by cycloheximide (CHX) chase (**Fig 1A**).

AUP1 was found to be associated with the HRD1 complex in mass spectrometry experiments [54]. However, published RNAi experiments had provided conflicting findings regarding a requirement for AUP1 in ERAD [35,39]. To establish whether, in our experimental

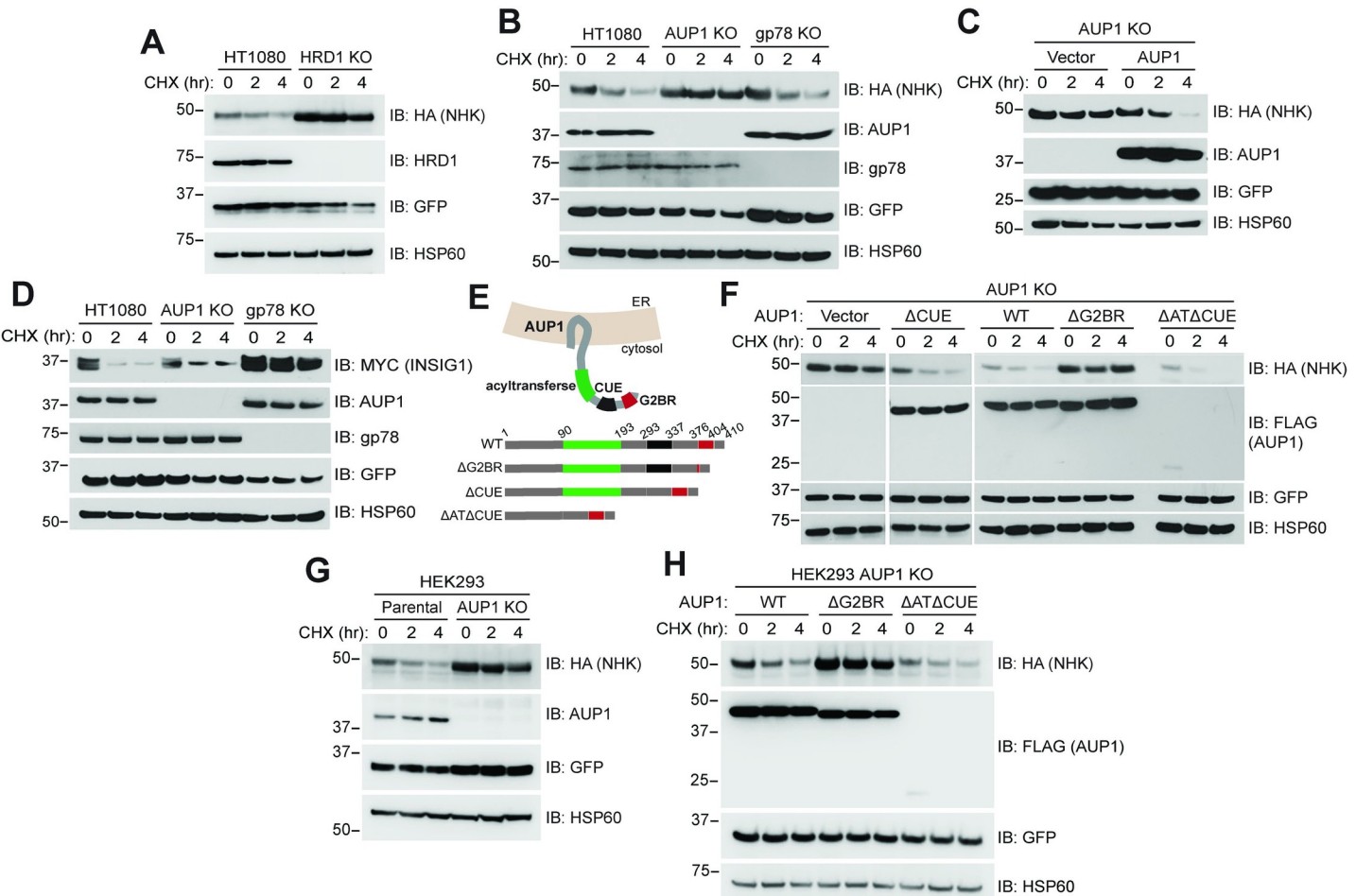

**Fig 1. The G2BR of AUP1 is required for ERAD of NHK. (A)** HT1080 or cells in which HRD1 was knocked out by CRISPR/Cas-9 (HRD1 KO) were transfected with plasmid encoding carboxyl-terminally HA-tagged NHK. Cells were treated with CHX for the indicated times to inhibit protein synthesis and NHK-HA degradation monitored by western blot. GFP and HSP60 serve as transfection efficiency and gel loading controls, respectively. **(B)** HT1080 or cells in which either AUP1 or gp78 was knocked out (AUP1 KO or gp78 KO) were transfected with NHK-HA and their degradation assessed. **(C)** AUP1 KO cells were transfected with NHK-HA and empty vector or plasmid encoding carboxyl-terminally FLAG-tagged AUP1 and NHK degradation assessed. **(D)** Indicated cells were transfected with plasmid encoding carboxyl-terminally MYC-tagged INSIG-1 and assessed for INSIG-1 degradation. **(E)** Schematic representation of AUP1 in the ER membrane and deletion mutants used in subsequent experiments. Amino acid residues are numbered spanning each domain. **(F)** AUP1 KO cells were transfected with NHK-HA and plasmids encoding the indicated FLAG-tagged AUP1 WT or deletion mutants and assessed for NHK degradation. **(G)** HEK293 (Parental) or HEK293 AUP1 KO cells were assessed for degradation of NHK-HA. **(H)** HEK293 AUP1 KO cells were assessed for NHK-HA degradation as in (F). Markers represent apparent molecular weight in kDa. The data underlying this figure can be found in **S1 Data**. aa, amino acid; AUP1, ancient ubiquitous protein 1; CHX, cycloheximide; ER, endoplasmic reticulum; ERAD, endoplasmic reticulum–associated degradation; G2BR, UBE2G2 Binding Region; KO, knockout; NHK, Null Hong Kong; WT, wild type.

system, AUP1 functions with HRD1 in the degradation of NHK, we utilized CRISPR/Cas-9 to generate an AUP1 KO in HT1080 cells (AUP1 KO). KO of AUP1 inhibited NHK degradation (**Fig 1B**), whereas a CRISPR/Cas-9 KO of the ERAD E3 gp78 had no effect on NHK turnover. Importantly, reexpression of full-length epitope-tagged AUP1 (AUP1-FLAG or wild type [WT]) restored NHK degradation in AUP1 KO cells (**Fig 1C**), ruling out possible clonal or off-target effects and definitively establishing a requirement for AUP1 in NHK turnover. These data confirm and extend previous findings on the effect of AUP1 loss in ERAD [35,40,41]. By contrast, degradation of INSIG-1, a known gp78 substrate [47,55], is dependent on gp78 and an intact G2BR$^{gp78}$ but was largely unaffected by loss of AUP1 (**Fig 1D, S1A Fig**). These findings establish the specificity of HRD1 and AUP1 in the targeting of NHK for

degradation and also establish a role for the G2BR$^{gp78}$ in the degradation of an ERAD substrate other than gp78 itself.

To determine the requirements for the multiple cytosolic domains of AUP1 in NHK degradation, we generated deletion mutants of FLAG-tagged AUP1 for expression in cells (see schematic **Fig 1E**). These included deletions of either the CUE domain (aa 293 to 337; ΔCUE), the acyltransferase plus the CUE domains (aa 90 to 337; ΔATΔCUE), or 12 of 26 aa within a region that is homologous to the G2BR of gp78 (aa 386 to 397; ΔG2BR). Each variant was expressed in the AUP1 KO cells and assessed for its ability to restore NHK degradation, with reexpression of full-length AUP1-FLAG (WT) serving as a positive control (**Fig 1F**). The ΔCUE mutant of AUP1-FLAG fully restored NHK degradation in the AUP1 KO cells. Similarly, and despite its rapid degradation and substantially lower steady state level (see **S1B Fig** for long exposure), the ΔATΔCUE mutant also restored NHK degradation. Quantitative PCR (qPCR) confirmed that the reduced expression of this mutant is due to posttranscriptional effects (**S1C Fig**). In contrast to the other deletions, the ΔG2BR mutant failed to restore NHK degradation, demonstrating the singular importance of the G2BR in the degradation of this substrate. This requirement for NHK degradation was recapitulated in HEK293 AUP1 KO cells (HEK293 AUP1 KO; **Fig 1G and 1H**).

We next assessed whether the specific requirement for the G2BR$^{AUP1}$ applies to another soluble ERAD-L substrate. For this, we examined the turnover of a well-characterized truncated mutant of Ribophorin I (RI$_{332}$) [35, 56]. Based on knockdown experiments both SEL-1L, which is a component of the HRD1 ligase complex, and AUP1 have been implicated in degradation of RI$_{332}$ [35,57]. Consistent with the SEL-1L observations, HRD1 KO cells exhibited a dramatic stabilization of RI$_{332}$ (**Fig 2A**), which was reversed by reexpression of HRD1. As with NHK, RI$_{332}$ degradation required the G2BR$^{AUP1}$ but was unaffected by loss of the acyltransferase and CUE domains (**Fig 2B**). Similar results were found in HEK293 AUP1 KO cells (**Fig 2C and 2D**).

To assess whether these requirements extend to substrates that engage the ERAD-C machinery, we followed the degradation of GFP$^u$, which is GFP rendered unstable in cells by fusion to the CL1 degron [45]. In human cells, degradation of CL1 fusion proteins has been reported to be dependent on AUP1 as well as on TRC8 and MARCH6 [40,41]. Loss of AUP1 prevented the rapid degradation of GFP$^u$ (**Fig 2E**), which again did not require either the CUE or the acyltransferase domains (**Fig 2F**). GFP$^u$ degradation was, however, completely abrogated by disruption of the G2BR$^{AUP1}$.

HRD1 is implicated in the basal degradation of 3-hydroxy-3-methylglutaryl CoA (HMG-CoA) reductase (HMGCR) [31]. In yeast, the targeting for degradation of the HMGCR ortholog, Hmg2p, by Hrd1p is an exemplar for the yeast ERAD-M pathway [8,58,59]. AUP1 has similarly been implicated in HMGCR degradation, although this has primarily been assessed in the context of its rapid degradation in response to high sterols [50], a process where at least four ERAD E3s have been reported to potentially play roles [22,55,60–62]. We have examined the basal degradation of endogenous HMGCR and find that it is also AUP1 dependent. As with the other substrates presented, HMGCR degradation uniquely required the G2BR$^{AUP1}$ (**S1D and S1E Fig**). Thus, of AUP1's three identified cytosolic domains, only the G2BR$^{AUP1}$ is required for ERAD-L substrates, a presumed ERAD-M substrate, as well as a substrate that engages the ERAD-C machinery.

## AUP1 binds specifically to UBE2G2

The G2BR$^{AUP1}$ sequence shares 41% identity to the G2BR of gp78 (G2BR$^{gp78}$), which binds to UBE2G2 but shows no discernable interaction with the most homologous mammalian E2,

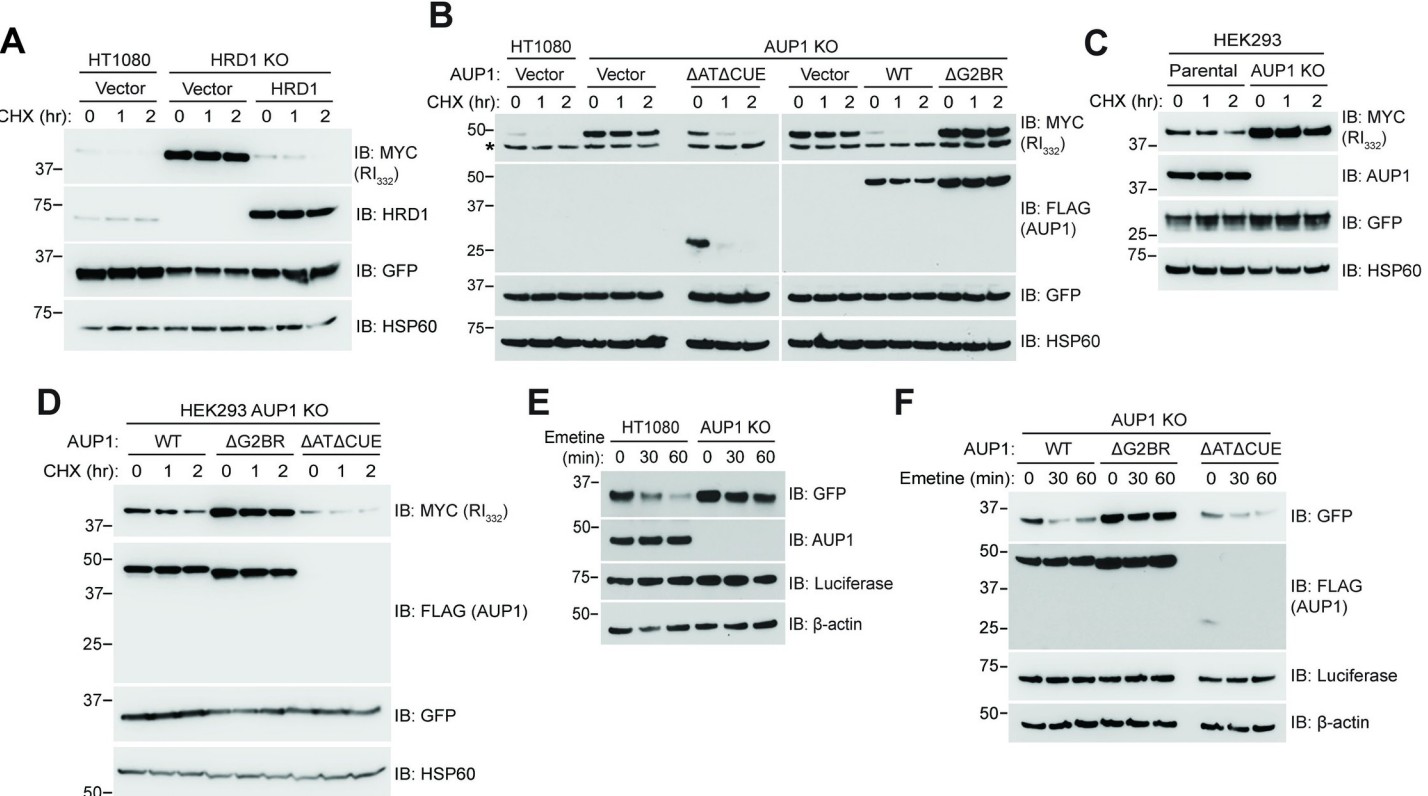

**Fig 2. The G2BR of AUP1 is required for ERAD of mutant ribophorin and GFP$^u$.** (A) HT1080 or HRD1 KO cells were transfected with plasmid encoding RI$_{332}$-MYC and with plasmid encoding HRD1 as indicated. Cells were subject to CHX chase and RI$_{332}$-MYC degradation monitored by western blot. GFP serves as a transfection efficiency control. (B) HT1080 or AUP1 KO cells were transfected with plasmid encoding RI$_{332}$-MYC and AUP1-FLAG WT or deletion mutants assessed for RI$_{332}$ degradation. Lower band (*) in MYC blot is nonspecific. (C, D) HEK293 or AUP1 KO cells were assessed for RI$_{332}$ turnover as in (B). (E) HT1080 or AUP1 KO cells were transfected with GFP$^u$ plasmid, and its degradation assessed by emetine chase, which has previously been used to assess GFP$^u$ turnover [40]. Luciferase and β-actin serve as transfection efficiency and gel loading controls, respectively. (F) HT1080 AUP1 KO cells were transfected with GFP$^u$ and the indicated AUP1-FLAG WT or deletion mutants, and GFP$^u$ degradation assessed as in (E). The data underlying this figure can be found in **S1 Data**. AUP1, ancient ubiquitous protein 1; CHX, cycloheximide; ERAD, endoplasmic reticulum–associated degradation; G2BR, UBE2G2 Binding Region; KO, knockout; WT, wild type.

UBE2G1 [63]. Consistent with this, a GST fusion of AUP1 that includes the G2BR bound UBE2G2 but not UBE2G1 (**Fig 3A**). Importantly, UBE2J1/UBC6e, which is an ER-localized transmembrane E2 that has been implicated in NHK degradation and suggested to function with AUP1 and other components of the HRD1 ERAD-L machinery [39,54,64], also failed to show detectable binding in vitro. UBE2D2/UBCH5B and UBE2D3/UBCH5C [65], which are considered promiscuous E2s and are involved in a number of different cellular processes including ERAD [40], similarly showed no discernable binding to G2BR$^{AUP1}$.

Having established the requirement for G2BR$^{AUP1}$ and the specificity of the interaction between G2BR$^{AUP1}$ and UBE2G2, NHK degradation was assessed in HT1080-derived cells in which UBE2G2 expression was abrogated using CRISPR/Cas-9 (UBE2G2 KO). As expected, INSIG-1, which is a substrate of gp78 and UBE2G2, was stable in these cells (**Fig 3B**). NHK was also stable in UBE2G2 KO cells (**Fig 3C**). Reexpression of UBE2G2 restored NHK degradation, while a catalytically inactive form of UBE2G2 (C89S) did not. These results establish a requirement for active UBE2G2 in the degradation of this HRD1 substrate. A similar requirement for UBE2G2 was established for the HRD1 substrate RI$_{332}$ (**S2A and S2B Fig**).

We next assessed ubiquitination of NHK in cells lacking either AUP1 or UBE2G2. Loss of either protein resulted in a marked decrease in substrate ubiquitination (**Fig 3D**), consistent

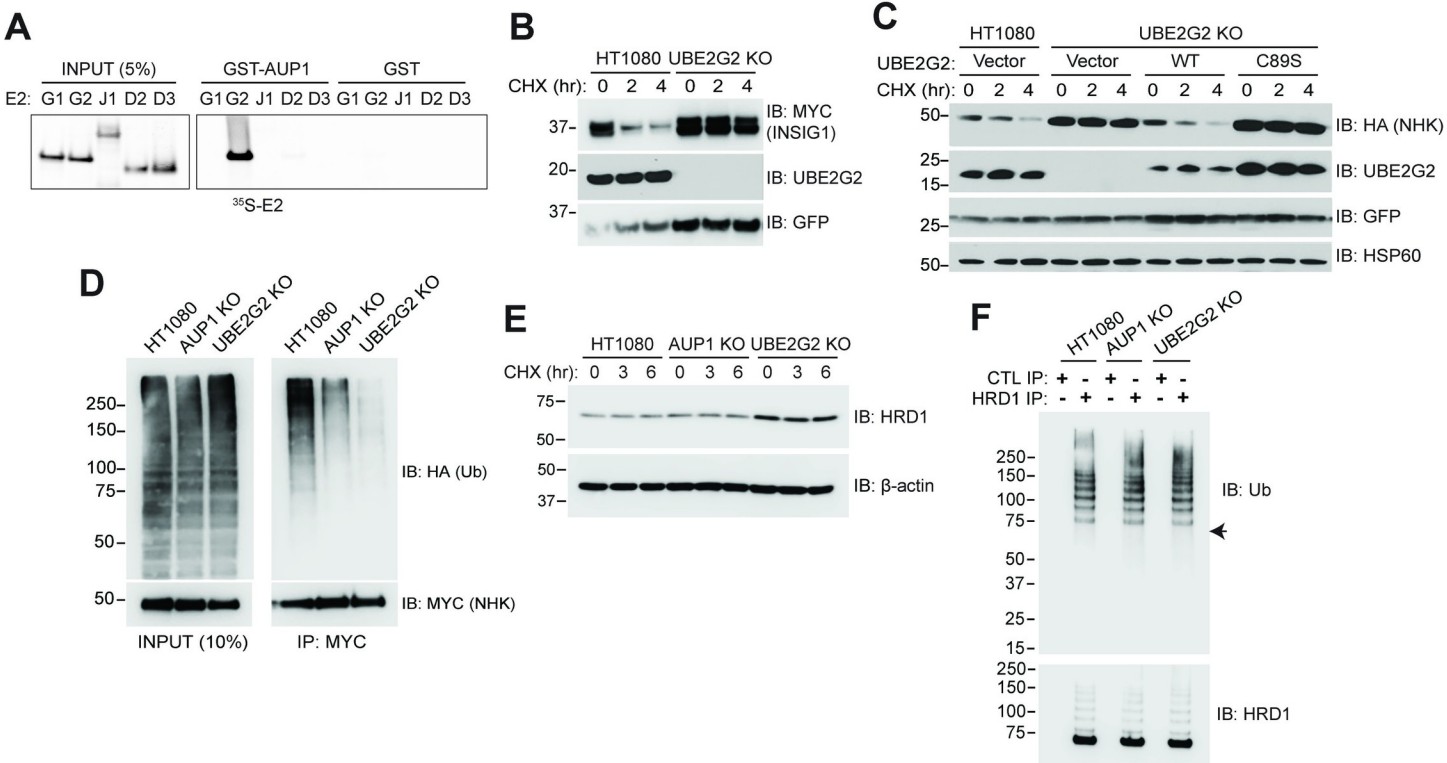

**Fig 3. The AUP1–UBE2G2 interaction is required for ERAD. (A)** Equimolar amounts of GST-AUP1 (aa 292 to 410) or GST were immobilized on glutathione beads and binding to the indicated in vitro translated $^{35}$S-labeled E2s was assessed. Left panel is 5% of input used for binding. E2s are referred to by their unique alpha-numeric designation (e.g., UBE2G1 and UBE2J1). **(B)** HT1080 or UBE2G2 KO cells were transfected with INSIG-1-MYC and degradation assessed by CHX chase. **(C)** The indicated cells were transfected with NHK-HA and empty vector, WT or catalytically inactive (C89S) N-terminally MYC-tagged UBE2G2, and NHK-HA degradation assessed. **(D)** Cells were transfected with NHK-MYC and HA-ubiquitin and ubiquitination of NHK assessed following proteasome inhibition and IP (right panel). Left panel is 10% of total cell lysate used for IP, indicating no discernable change in total ubiquitination. **(E)** Levels of endogenous HRD1 were monitored by CHX chase in the indicated cell types. **(F)** Ubiquitination of endogenous HRD1 was assessed following IP with HRD1 antiserum. Arrow in upper panel indicates migration of unmodified HRD1. Control immunoprecipitates using rabbit serum are shown for comparison. The data underlying this figure can be found in **S1 Data**. aa, amino acid; AUP1, ancient ubiquitous protein 1; CHX, cycloheximide; ERAD, endoplasmic reticulum–associated degradation; IP, immunoprecipitation; KO, knockout; WT, wild type.

with reduced NHK turnover in both AUP1 and UBE2G2 KO cells. This diminution of ubiquitination is not due to a reduction in the levels of HRD1 (**Fig 3E**). If anything, the steady state level of this E3 is increased in UBE2G2 KO cells. This increase in HRD1 is consistent with the induction of an ER stress response in these cells, which lack a critical component of the ERAD machinery (**S2C and S2D Fig**). In yeast, there is evidence that ubiquitination of Hrd1p itself is critical to its role in substrate retrotranslocation and degradation [66–68]. We found that in HT1080 cells, ubiquitination of HRD1 is readily detectable and that there is no apparent alteration in this ubiquitination with loss of either AUP1 or UBE2G2 (**Fig 3F, S2E and S2F Fig**). Despite its robust ubiquitination, there is no evidence that HRD1 is degraded over a period of 6 hours (**Fig 3E**). This suggests that this AUP1 and UBE2G2-independent ubiquitination does not target HRD1 for degradation, whereas AUP1- and UBE2G2-mediated ubiquitination is required for substrate degradation.

## Structural and biophysical characterization of UBE2G2:G2BR$^{AUP1}$

We have determined the crystal structure of UBE2G2 in complex with G2BR$^{AUP1}$ at 1.74-Å resolution (PDB entry 7LEW, **Table 1**). The UBE2G2:G2BR$^{AUP1}$ structure reveals UBE2G2 residues 2 to 96 and 107 to 165, G2BR$^{AUP1}$ residues 377 to 404, and 158 oxygen atoms of water

**Table 1. Data collection and refinement statistics[a].**

| DATA COLLECTION | |
|---|---|
| Space group | $P2_12_12_1$ |
| Cell dimensions | |
| $a, b, c$ (Å) | 49.74, 58.23, 63.39 |
| $\alpha, \beta, \gamma$ (°) | 90, 90, 90 |
| Resolution (Å) | 50 to 1.74 (1.80 to 1.74) |
| Number of unique reflections | 19,573 (1,917) |
| $R_{merge}$ (%)[b] | 3.4 (50.2) |
| $I/\sigma$ | 42.4 (3.8) |
| Completeness (%) | 99.6 (99.5) |
| Redundancy | 5.4 (5.6) |
| **REFINEMENT** | |
| Resolution | 37.8 to 1.74 (1.83 to 1.74) |
| $R_{work}$ (%)[c] | 19.15 (24.77) |
| $R_{free}$ (%)[d] | 22.79 (30.52) |
| No. of atoms | |
| Protein | 1,628 |
| Water | 158 |
| $B$ factors | |
| Protein | 33.02 |
| Water | 38.46 |
| RMSD | |
| Bond lengths (Å) | 0.011 |
| Bond angles (°) | 1.116 |
| Ramachandran plot | |
| In preferred regions (%) | 95.8 |
| In allowed regions (%) | 4.2 |
| Outliers (%) | 0 |

[a]Values in parentheses are for the highest resolution shell.

[b]$R_{merge} = \Sigma|(I - <I>)|/\sigma(I)$, where $I$ is the observed intensity.

[c]$R_{work} = \Sigma_{hkl} | |F_o| - |F_c| | / \Sigma_{hkl} |F_o|$, calculated from working data set.

[d]$R_{free}$ is calculated from approximately 1,000 reflections randomly chosen and not included in refinement.

RMSD, root–mean–square deviation.

molecules. Residue 1 and residues 97 to 106 of the extended loop characteristic of UBE2G1 and UBE2G2, as well as yeast Ubc7p, are disordered without observable electron density. The overall structure exhibits a high degree of similarity to the previously described structure of UBE2G2:G2BR[gp78] [17, 30] with a root–mean–square deviation (RMSD) of 0.5 Å for 167 out of 182 pairs of Cα atoms. This RMSD remains for 145 out of 154 pairs of UBE2G2 Cα atoms. **Fig 4A** depicts virtually identical ribbon depictions of the two G2BR helices resulting from the superposition of the two UBE2G2 Cα traces.

The UBE2G2:G2BR complex is stabilized with two types of interactions across the interface. One type includes hydrogen bonds and salt bridges, electrostatic in nature (**Fig 4B**), and the other type includes van der Waals contacts, i.e., hydrophobic interactions. The high degree of similarity of the UBE2G2:G2BR[AUP1] and UBE2G2:G2BR[gp78] structures is the result of a largely conserved set of G2BR residues involved in direct contacts with UBE2G2 (**Fig 4C**). It is not surprising that this set of residues in the two G2BR sequences forms an almost identical

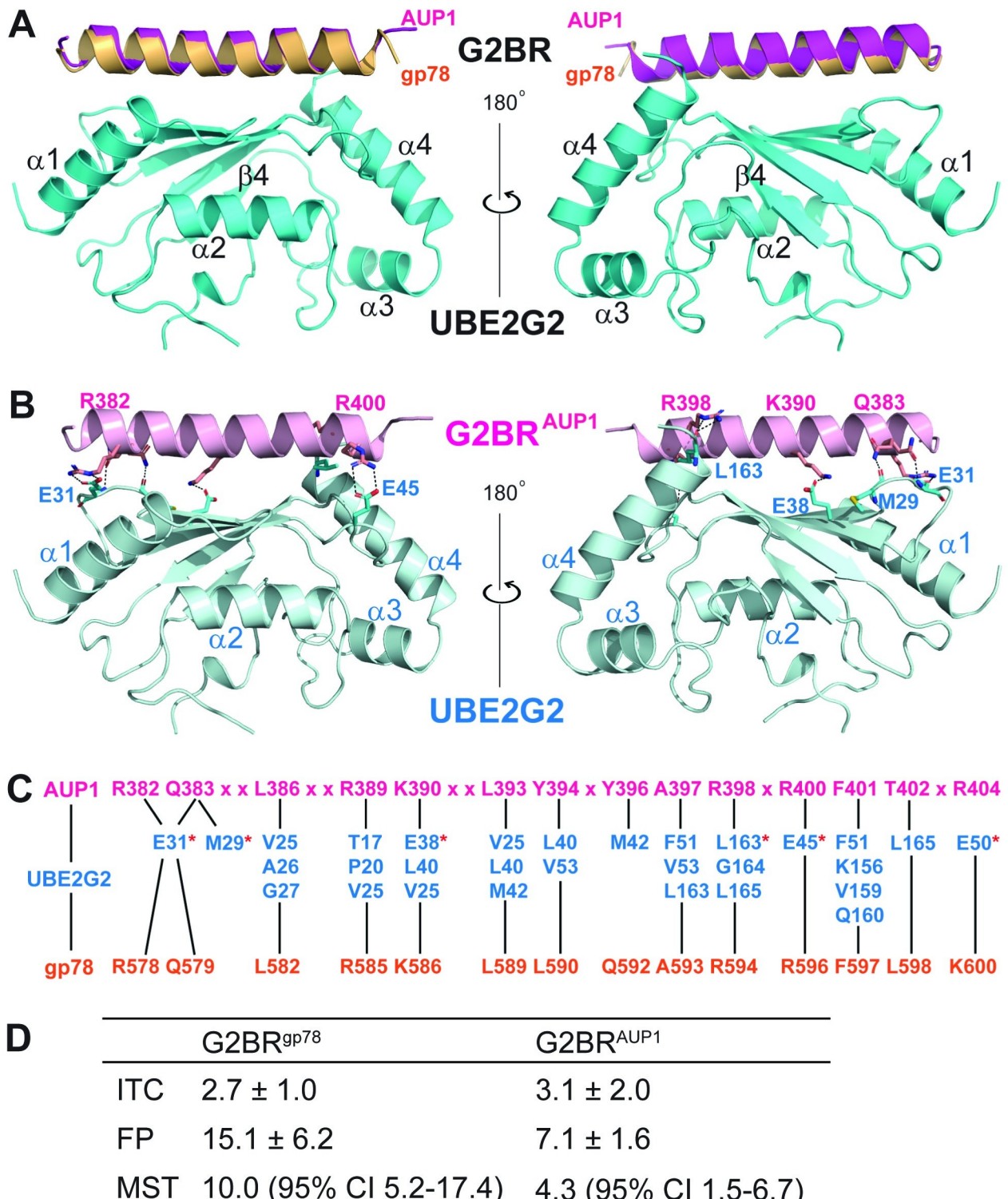

**Fig 4. Crystal structure of the UBE2G2:G2BR^AUP1 complex. (A)** Ribbon representation of the superimposed UBE2G2:G2BR^AUP1 (in cyan and magenta, this work) and UBE2G2:G2BR^gp78 (in orange, PDB entry 3H8K) complexes. The UBE2G2 in complex with G2BR^gp78 is not shown for high similarities between the two structures. The left and right views are related by a 180˚ rotation around the vertical axis. **(B)** Intermolecular hydrogen bonds and salt bridges between UBE2G2 and G2BR^AUP1 are shown as dotted lines. Involved aa residues are shown as stick models in atomic color scheme (N in blue, C in cyan or magenta, O in red, and S in orange). The left and right views are related by a 180˚ rotation around the vertical axis. **(C)** Contacts between UBE2G2 (in cyan) and G2BR^AUP1 (in magenta) are conserved in the UBE2G2:G2BR^gp78 (in orange) complex. The UBE2G2

                                           

residues involved in hydrogen bonds and/or salt bridges are indicated with a red star. Black lines link each G2BR residue to its reciprocal contact(s) in UBE2G2, and "x" denotes G2BR[AUP1] residues that do not contact UBE2G2 directly. (**D**) Calculated dissociation constants ($K_d$, in nM) of G2BR peptides for UBE2G2 from ITC, FP, or MST, experiments. Mean and standard deviations or 95% CI for at least two titrations are reported. aa, amino acid; CI, confidence interval; FP, fluorescence polarization; G2BR, UBE2G2 Binding Region; ITC, isothermal titration calorimetry; MST, microscale thermophoresis.

pattern of interactions with the E2 molecule in the UBE2G2:G2BR[AUP1] and UBE2G2: G2BR[gp78] complexes.

To quantify the affinity between UBE2G2 and G2BR[AUP1], we performed isothermal titration calorimetry (ITC) measurements with purified UBE2G2 and either the G2BR[gp78] or the G2BR[AUP1] peptide. This resulted in measured $K_d$ values of 2.7 ± 1.0 nM and 3.1 ± 2.0 nM, respectively (**Fig 4D, S3A Fig**), which is in accordance with the previously reported high-affinity interaction between UBE2G2 and G2BR[gp78] [17,30]. We also utilized both microscale thermophoresis (MST) and fluorescence polarization (FP) assays with purified UBE2G2 and fluorescently labeled G2BR[gp78] or G2BR[AUP1] peptides. Both methods confirmed the low nanomolar $K_d$ values (**Fig 4D, S3B and S3C Fig**). Based on these results, we conclude that the affinity of G2BR[AUP1] for UBE2G2 is similar to that of G2BR[gp78].

## The G2BR[AUP1] allosterically increases the affinity of UBE2G2 for the gp78 RING domain

Our previous studies of UBE2G2:G2BR[gp78] revealed a significant change in nuclear magnetic resonance (NMR) chemical shifts for the backbone $^{15}$N-$^1$H$^N$ resonances of UBE2G2 upon binding of G2BR[gp78] [17,24]. The recognition that the backbone structure of UBE2G2 changes very little between the apo and G2BR[gp78] bound states suggested that the chemical shifts were due to variations in the hydrogen bond strengths that stabilize the core structure. Nevertheless, subtle changes in populations of dynamic conformers at the gp78 RING:UBE2G2 interface take place to promote the significant allosteric effect observed for binding of gp78 RING to UBE2G2 when G2BR[gp78] is bound [69]. We explored whether the same effects are present in the interaction with G2BR[AUP1]. The $^{15}$N-$^1$H$^N$ chemical shifts of UBE2G2 exhibit the same dramatic shifts upon binding G2BR[AUP1] as upon binding G2BR[gp78], suggesting similar allosteric effects (**Fig 5A–5C**, Biological Magnetic Resonance Data Bank deposition https://doi.org/10.13018/bmrbig33). Furthermore, consistent with the very similar crystal structures of UBE2G2: G2BR[gp78] and UBE2G2:G2BR[AUP1] that we report here, the UBE2G2 chemical shifts in the $^{15}$N-$^1$H$^N$ HSQC spectrum showed a very similar shift from the apo UBE2G2 (**Fig 5C**).

We have previously found that the addition of G2BR[gp78] or U7BR to their cognate E2s resulted in a significant increase in E2:RING affinity as assessed by NMR, which correlated with enhanced ubiquitination in vitro [17,19]. Taking advantage of the well-characterized gp78 RING domain, we assessed the relative effects of G2BR[gp78] and G2BR[AUP1] on the affinity of UBE2G2 for this domain. Titration of UBE2G2 and UBE2G2:G2BR[AUP1] with the gp78 RING confirmed that the gp78 RING exhibited mostly fast exchange kinetics and that the increased affinity of gp78 RING was enhanced to a similar degree by G2BR[AUP1] as by G2BR[gp78]. Through chemical shift perturbation (CSP) mapping, backbone amide resonances of G2BR-bound UBE2G2 were examined upon titration of the RING domain (**Fig 5D and 5E**, Biological Magnetic Resonance Data Bank deposition https://doi.org/10.13018/bmrbig33). Titration of UBE2G2:G2BR[gp78] and of UBE2G2:G2BR[AUP1] resulted in measured affinities of 20.3 ± 1.5 μM and 22.5 ± 1.8 μM, respectively. These results indicate equivalent enhancements of approximately 10-fold for G2BR[gp78] and G2BR[AUP1] on the affinity of UBE2G2 for the gp78 RING, which was measured as 207.4 ± 6.4 μM. Consistent with this increase in affinity, the

                          

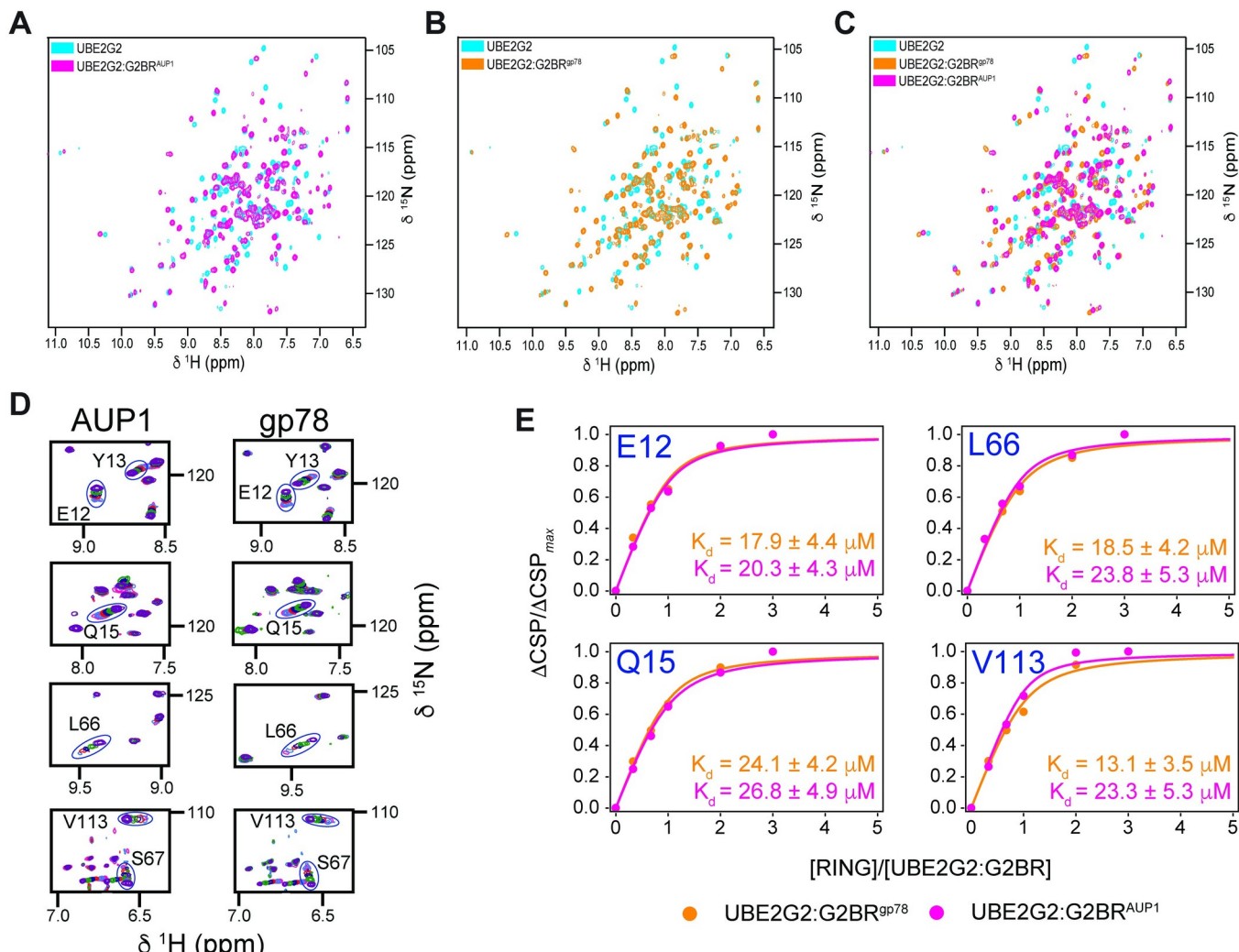

**Fig 5. Titration of UBE2G2:G2BR with gp78 RING.** Superposition of $^{15}N$-$^1H^N$ HSQC NMR spectra for apo-UBE2G2 and UBE2G2 bound to G2BR$^{AUP1}$ **(A)** or G2BR$^{gp78}$ **(B)**. Spectra for UBE2G2 (150 µM) bound to G2BR were acquired at a 1:1.1 ratio. **(C)** Superposition of the spectra for apo-UBE2G2, UBE2G2: G2BR$^{AUP1}$, and UBE2G2:G2BR$^{gp78}$, illustrating the equivalence of perturbations between G2BR$^{AUP1}$ and G2BR$^{gp78}$. **(D)** Expansion of overlaid resonances corresponding to UBE2G2 residues showing fast exchange binding kinetics for gp78 RING binding to both UBE2G2:G2BR$^{AUP1}$ (left panels) and UBE2G2: G2BR$^{gp78}$ (right panels). Spectra were acquired at ratios of RING to UBE2G2:G2BR of 0, 0.33, 0.67, 1.0, 2.0, and 3.0. **(E)** Binding curves as a function of [RING]/[UBE2G2:G2BR] ratio and analyses for representative residues E12, Q15, L66, and V113 of UBE2G2:G2BR$^{gp78}$ (orange) and UBE2G2:G2BR$^{AUP1}$ (magenta) UBE2G2:G2BR binding to gp78 RING. Errors are reported from the regression analysis of the data in **S1 Table** (B, D). G2BR, UBE2G2 Binding Region; NMR, nuclear magnetic resonance.

G2BR$^{AUP1}$, like the G2BR$^{gp78}$, increased ubiquitination in an in vitro autoubiquitination assay employing UBE2G2 with either the gp78 RING domain or those of HRD1 and TRC8 (**S4A and S4B Fig**). Thus, as assessed both biophysically and functionally, G2BR$^{AUP1}$ has similar effects on UBE2G2 activity to what is observed with G2BR$^{gp78}$.

## The AUP1–UBE2G2 interaction is required for ERAD

Inspection of the UBE2G2:G2BR$^{AUP1}$ interface reveals several contact residues that are likely responsible for their high-affinity binding. Specifically, five positively charged or polar residues of the G2BR$^{AUP1}$—R382, Q383, K390, R398, and R400—form salt bridges or hydrogen bonds with negatively charged or polar groups on UBE2G2 (**Fig 4B and 4C**). We first generated

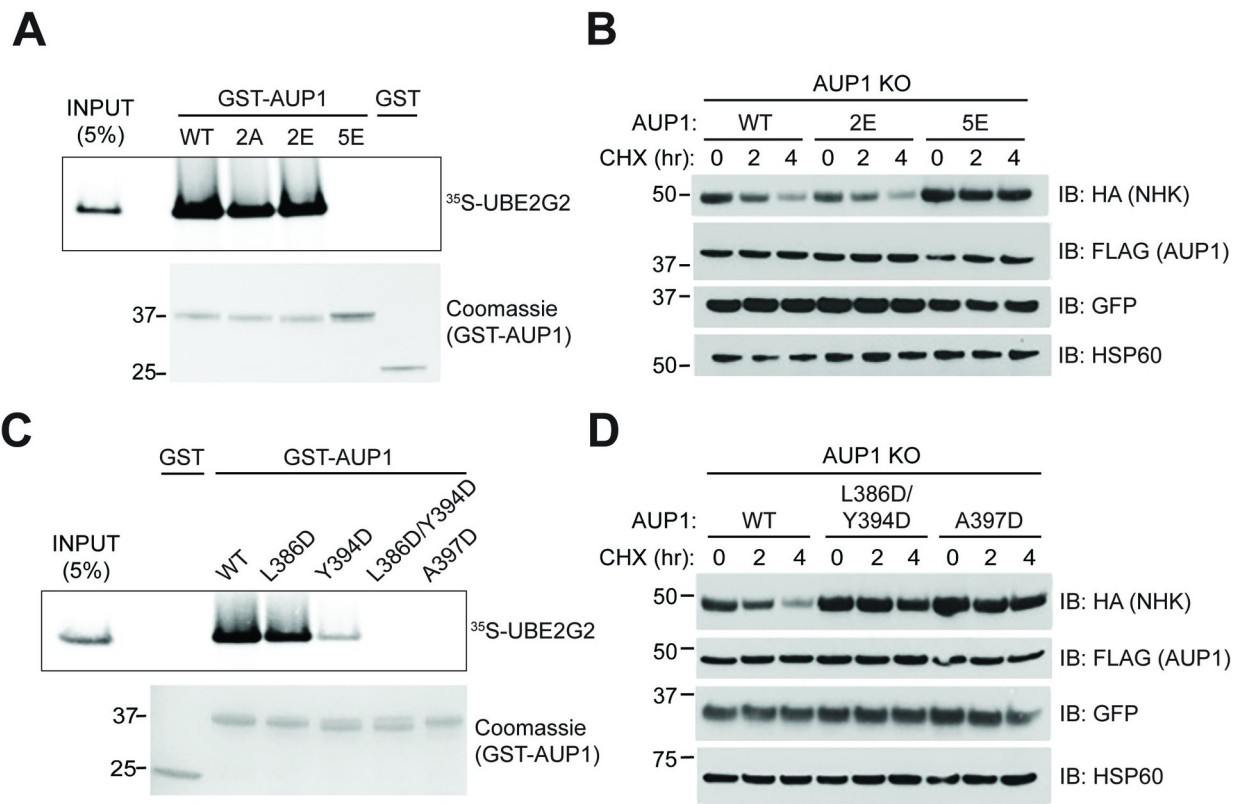

**Fig 6. Disruption of UBE2G2:G2BR<sup>AUP1</sup> interaction abrogates AUP1 function in cells.** (A) GST or indicated GST-AUP1 fusions (aa 292 to 410, expression shown in Coomassie-stained gel) were immobilized on glutathione beads and binding to $^{35}$S-UBE2G2 was assessed as in Fig 3A. Mutations include 2A (R382A/Q383A), 2E (R382E/Q383E), and 5E (R382E/Q383E/K390E/R398E/R400E). (B) AUP1 KO cells were transfected with NHK-HA and forms of full-length AUP1-FLAG with the indicated point mutations from (A) and NHK degradation assessed by CHX chase. (C) Binding assay performed as in (A) with AUP1 mutants predicted to disrupt hydrophobic interactions with UBE2G2. (D) Cellular assay performed as in (B) with AUP1-FLAG mutants that disrupt interactions with UBE2G2 in vitro in (C). The data underlying this figure can be found in S1 Data. aa, amino acid; AUP1, ancient ubiquitous protein 1; CHX, cycloheximide; G2BR, UBE2G2 Binding Region; KO, knockout; WT, wild type.

alanine mutations on these G2BR<sup>AUP1</sup> residues to neutralize positively charged side chains and assessed their effects on UBE2G2 binding and activity in cells by GST pulldown and CHX chase, respectively. Beginning at the N-terminal end of the G2BR<sup>AUP1</sup> helix, the R382A/Q383A mutation (2A) was unable to disrupt UBE2G2 binding (**Fig 6A**). Charge reversal of these residues (R382E/R383E, or 2E) also failed to disrupt UBE2G2 binding. Mutation of all five of the positively charged or polar G2BR<sup>AUP1</sup> residues to glutamate (5E), however, led to a loss of detectable UBE2G2 binding. Consistent with the importance of binding to UBE2G2, expression of a full-length AUP1 bearing the 2E mutation restored degradation of NHK in the AUP1 KO cells, to a degree comparable to WT AUP1 (**Fig 6B**). By contrast, expression of the AUP1 5E mutant failed to restore NHK degradation in these cells. These results support a requirement for UBE2G2 binding in AUP1 function in cells.

As mentioned, the pattern of hydrophobic interactions in the UBE2G2:G2BR<sup>AUP1</sup> and UBE2G2:G2BR<sup>gp78</sup> complexes are highly similar (**Fig 4C**), suggesting that the van der Waals forces resulting from a perfect landscape match of the 2 molecules is also required for the stabilization of the complex. To test this hypothesis, we mutated the G2BR<sup>AUP1</sup> residues L386, Y394, and A397 to aspartate to disrupt this hydrophobic landscape and assessed effects on in vitro binding of UBE2G2. While the Y394D mutation alone reduced binding, mutation of

L386 and Y394 together eliminated detectable binding to UBE2G2 (**Fig 6C**). Binding was similarly lost with a single point mutation, A397D, which makes multiple hydrophobic contacts with UBE2G2 (**Fig 4C**). Consistent with the in vitro binding results, reexpression of AUP1 harboring either the L386D/Y394D or A397D mutations failed to restore NHK degradation in cells lacking AUP1, confirming the functional importance of these residues (**Fig 6D**). Taken together, our mutational analysis demonstrates that the high-affinity binding between G2BR$^{AUP1}$ and UBE2G2, driven by both electrostatic and hydrophobic interactions, is critical for the function of AUP1 in ERAD.

## The G2BR protects UBE2G2 from degradation

Previous studies have demonstrated that yeast Cue1p protein and its U7BR, which binds its E2 in an overall similar manner as the G2BR, protects Ubc7p from a degradative process that is dependent on the activity of the E2 [18,19,70]. The exact mechanism by which Cue1p protects Ubc7p remains unclear. Strikingly, despite the continued expression of gp78 and its G2BR, HT1080 cells lacking AUP1 showed substantially reduced levels and increased turnover of UBE2G2 (**Fig 7A**). On the other hand, cells lacking gp78 expression maintained normal levels of endogenous UBE2G2. Transcript levels of UBE2G2 were not affected by loss of AUP1 (**Fig 7B**). By transfecting either UBE2G2 or the catalytically inactive C89S mutant, we established that the observed turnover of UBE2G2 requires its E2 activity (**Fig 7C**). Increased distribution of AUP1 from the ER to lipid droplets has been reported in cells loaded with oleic acid [35]. This raises the interesting possibility that accumulation of lipid droplets in cells, such as observed in obesity, could affect ERAD by altering the availability or level of UBE2G2. However, in our system, we find that loading HT1080 cells with oleic acid, and thereby increasing lipid droplets, affected neither UBE2G2 levels nor the degradation of NHK (**S5A and S5B Fig**).

To determine whether the G2BR$^{AUP1}$ is responsible for preventing UBE2G2 from undergoing constitutive degradation, we generated constructs encoding GFP fusions with either G2BR$^{AUP1}$ or G2BR$^{gp78}$ and transfected these into AUP1 KO cells. When compared to the control (a scrambled (SCR) G2BR sequence fused to GFP), either GFP-G2BR$^{AUP1}$ or GFP-G2BR$^{gp78}$ protected UBE2G2 from degradation and increased the steady state level of this E2 (**Fig 7D**; see **S5C Fig** for quantification of representative GFP-G2BR overexpression). A similar loss of UBE2G2 and rescue by GFP-G2BR overexpression was observed in HEK293 AUP1 KO cells (**S5D and S5E Fig**), and a dependency of E2 stability on AUP1 was also observed in M17, a neuroblastoma cell line (**S5D Fig**). The question then arises as to why endogenous gp78 is inadequate to maintain UBE2G2 levels in the absence of AUP1. A possible explanation could be the relative amounts of the two G2BR-containing proteins. By making use of existing AUP1 and gp78 antibodies and GST fusion proteins, we estimate that there is nearly 40-fold more AUP1 relative to gp78 in HT1080 cells (**Fig 7E**). This difference in prevalence between AUP1 and gp78 and, by extension, their G2BRs, provides at least one plausible explanation for the differential role of AUP1 in protecting UBE2G2. A substantially higher level of expression of AUP1 relative to gp78 is also observed in HEK293 and other human cell lines that we evaluated (**S5F Fig**). While these findings collectively suggest why AUP1 is important for maintaining UBE2G2 levels, the molecular basis for G2BR-mediated protection of UBE2G2 remains to be determined.

## UBE2G2 membrane recruitment and activation by G2BR are necessary for ERAD

One simple explanation for the requirement for AUP1 in degradation of HRD1 substrates is that the dramatic reduction in UBE2G2 in the absence of AUP1 prevents efficient ERAD.

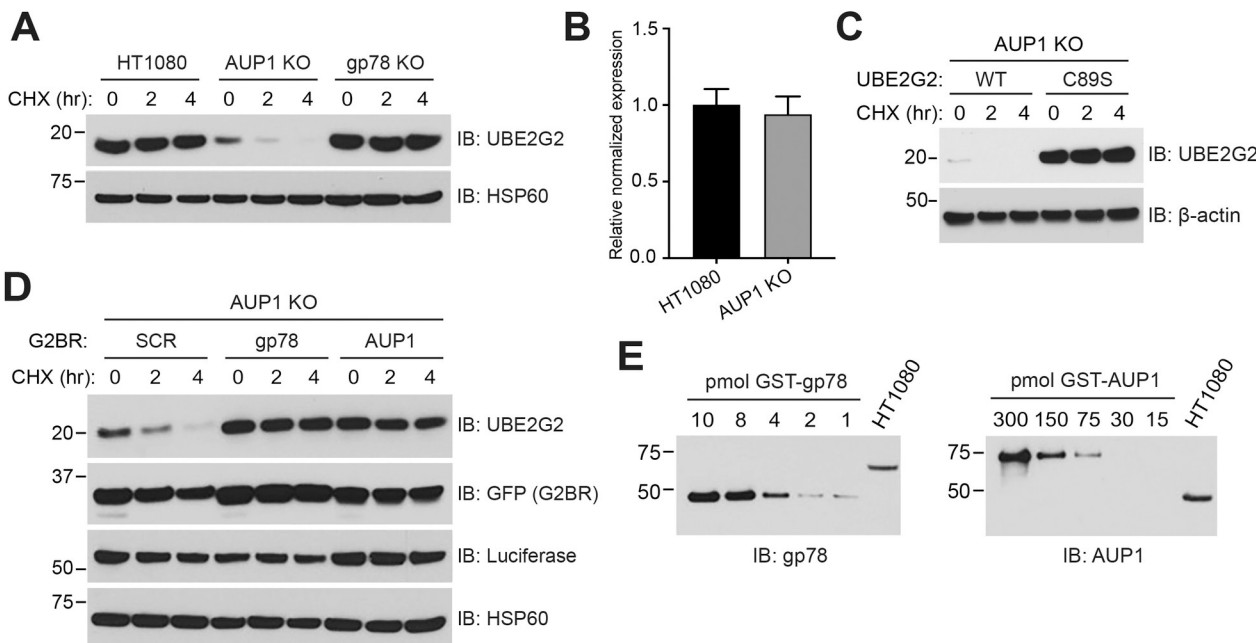

**Fig 7. G2BR^AUP1 protects UBE2G2 from degradation. (A)** Indicated cells were assessed for steady state levels and turnover of endogenous UBE2G2 by CHX chase. **(B)** Transcript levels of endogenous UBE2G2 in AUP1 KO cells compared to HT1080 as determined by qPCR. Mean and standard deviation from three independent experiments is shown. **(C)** Turnover of transfected MYC-UBE2G2 or a catalytically inactive mutant (C89S) was monitored in AUP1 KO cells. **(D)** AUP1 KO cells were transfected with plasmids encoding MYC-UBE2G2 and GFP fusions of either a scrambled G2BR (SCR), G2BR^gp78, or G2BR^AUP1, and the turnover of UBE2G2 was assessed. **(E)** Relative abundance of gp78 (left panel) and AUP1 (right panel) in HT1080 cells was determined by resolving indicated amounts (in pmoles) of purified GST-gp78 (aa 429 to 611) or GST-AUP1 (aa 61 to 410) fusions alongside approximately $1 \times 10^4$ cellular equivalents of cell lysate and immunoblotting with gp78 or AUP1 antibodies. For each antibody, the amount of immunoreactivity in the lysate was compared to a defined quantity of a GST fusion protein of gp78 or AUP1 spanning the epitope recognized by the probing antibody. This analysis suggests there is close to 40-fold as much cellular AUP1 (150 pmole) as gp78 (4 pmol). The data underlying this figure can be found in **S1 and S8 Data**. aa, amino acid; AUP1, ancient ubiquitous protein 1; CHX, cycloheximide; G2BR, UBE2G2 Binding Region; KO, knockout; qPCR, quantitative polymerase chain reaction; WT, wild type.

However, as already shown, the degradation of INSIG-1, an established gp78 substrate that requires UBE2G2 (**Fig 3B**), was largely unaffected by AUP1 loss (**Fig 1D**). Even so, perhaps UBE2G2 levels are too low to support NHK degradation in the absence of AUP1. If this were the case, then simply overexpressing UBE2G2 might be expected to correct this deficit and restore NHK degradation. However, overexpression of MYC-tagged UBE2G2 to levels exceeding that of the endogenous protein did not restore NHK degradation in AUP1 KO cells (**Fig 8A**). This outcome establishes a requirement for AUP1 for ERAD function, beyond maintaining UBE2G2 levels. Notably, as seen also in **Fig 7D**, the overexpressed E2 was also unstable in AUP1 KO cells, although it is clearly present at higher levels than the endogenous E2 (**Fig 8A**).

Based on the specific, high-affinity interaction we observe between UBE2G2 and G2BR^AUP1, we asked whether expression of G2BR^AUP1 as a GFP fusion is sufficient to both increase UBE2G2 levels and restore ERAD in AUP1 KO cells. Overexpression of GFP-G2BR^AUP1, despite protecting UBE2G2 from degradation and increasing the levels of endogenous UBE2G2, was insufficient to reconstitute NHK degradation (**Fig 8B**). Expression of an AUP1 construct lacking only the hairpin membrane anchor was similarly unable to restore NHK degradation (**S6A Fig**). We next overexpressed UBE2G2 and GFP-G2BR^AUP1 together. Again, this did not restore NHK degradation, despite supraphysiological levels of UBE2G2 (**Fig 8C**; for a direct comparison of the level of endogenous UBE2G2 in HT1080 cells to the levels of transfected UBE2G2 levels at the zero time points, see **S6B Fig, lanes 1 to 3**).

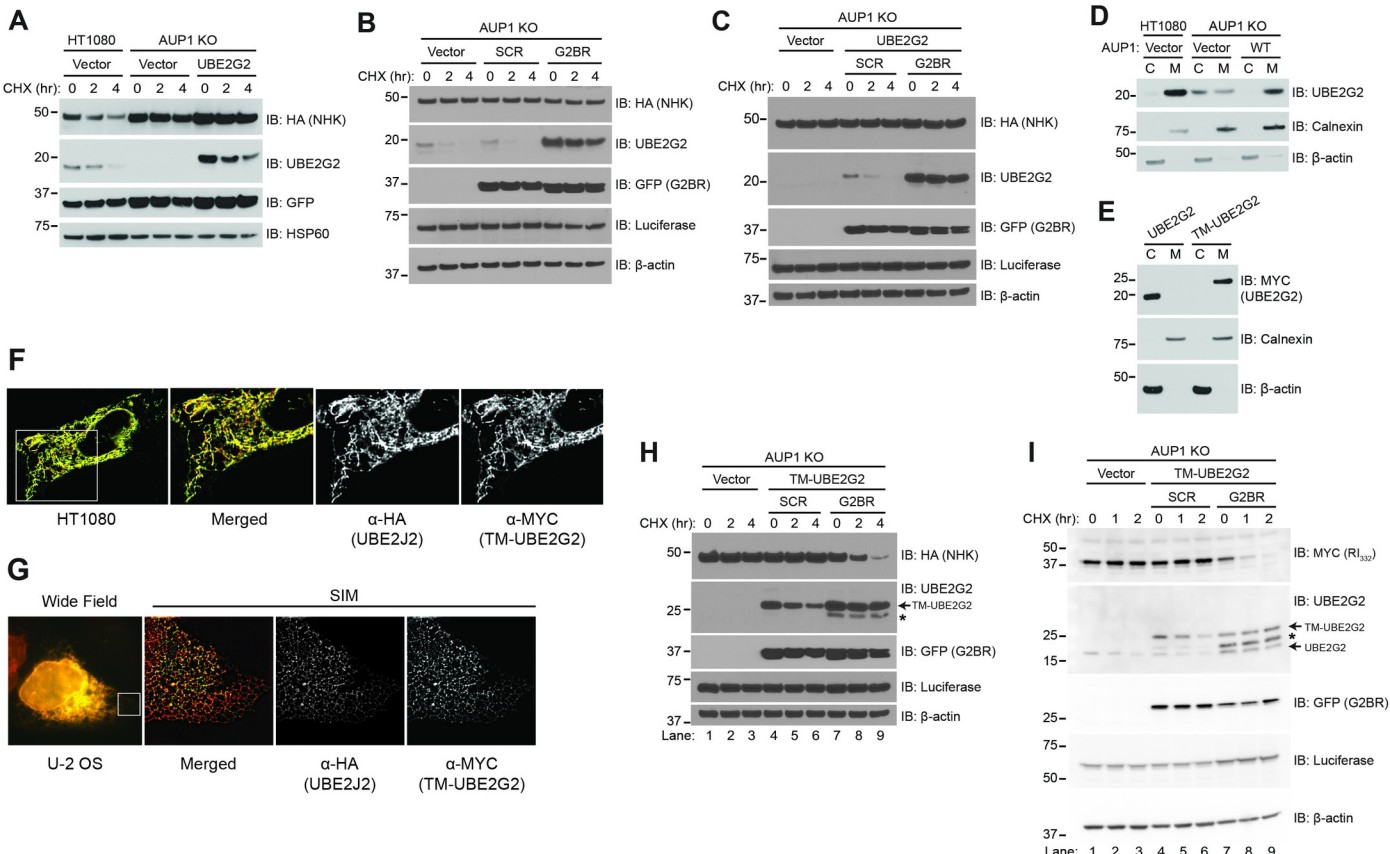

**Fig 8. Recruitment of UBE2G2 to the ER membrane and activation by G2BR^AUP1 is required for ERAD. (A)** HT1080 or AUP1 KO cells were transfected with NHK-HA and either empty vector or MYC-UBE2G2 and NHK turnover assessed by CHX chase. **(B)** AUP1 KO cells were transfected with NHK-HA and GFP fusions of either a scrambled G2BR sequence (SCR) or G2BR^AUP1 and NHK turnover and levels of endogenous UBE2G2 assessed. Luciferase and β-actin served as transfection efficiency and gel loading controls, respectively. **(C)** AUP1 KO cells were transfected with NHK-HA, MYC-UBE2G2, and either GFP-SCR or GFP-G2BR^AUP1 and NHK turnover assessed. **(D)** Membrane fractionation was performed on HT1080 or AUP1 KO cells transfected with either empty vector or AUP1-FLAG and localization of UBE2G2 assessed by western blotting. Localization of β-actin (cytosol) and Calnexin (ER membrane) were used as controls for cytosolic (C) and membrane (M) fractions, respectively. **(E)** Membrane fractionation of AUP1 KO cells transfected with either MYC-UBE2G2 or TM-UBE2G2. **(F)** HT1080 cells were transfected with TM-UBE2G2 and its localization assessed by confocal microscopy. Images shown are from an optical section after deconvolution. HA-tagged UBE2J2, an ER-localized E2, serves as an ER marker. **(G)** U-2 OS cells were transfected as in (F), and localization assessed by SIM. **(H)** AUP1 KO cells were transfected with NHK-HA and TM-UBE2G2 and either GFP-SCR or GFP-G2BR^AUP1 and NHK turnover assessed. TM-UBE2G2 is indicated by the arrow, an incomplete form is indicated by *. **(I)** AUP1 KO cells were transfected with RI_332-MYC, TM-UBE2G2, and either GFP-SCR or GFP-G2BR^AUP1 and RI_332 turnover assessed. Endogenous UBE2G2 and TM-UBE2G2 are indicated, as is the incomplete form of the latter (*). The data underlying this figure can be found in **S1 Data**. AUP1, ancient ubiquitous protein 1; CHX, cycloheximide; ER, endoplasmic reticulum; ERAD, endoplasmic reticulum–associated degradation; G2BR, UBE2G2 Binding Region; KO, knockout; SIM, structured illumination microscopy.

We have shown that an internal deletion mutant of AUP1, which lacks both the acyltransferase and CUE domains but retains its membrane anchor and the G2BR, restores NHK degradation in AUP1 KO cells (**Fig 1F**). Therefore, it is possible that, in the context of the substrates being analyzed, the primary function of AUP1 is to recruit UBE2G2 to the ER membrane. In support of a role for AUP1 in ER membrane recruitment, UBE2G2 was overwhelmingly found in the membrane fraction in both WT HT1080 cells and in AUP1 KO cells when AUP1 was reexpressed (**Fig 8D**). By contrast, UBE2G2 was distributed between the membrane and cytosolic fractions in the AUP1 KO cells. To address the importance of G2BR^AUP1 in recruiting UBE2G2 to the ER membrane, we fused MYC-tagged UBE2G2 to the transmembrane domain of the yeast ERAD accessory protein Cue1p [14] to generate TM-UBE2G2. This transmembrane domain has been previously used to localize the cytoplasmic region of gp78 to the ER,

which was found to be sufficient to restore degradation of gp78 substrates in cells where gp78 has been knocked down [28]. TM-UBE2G2 is inserted into membranes as assessed by fractionation (**Fig 8E**) and localizes to the ER as revealed by fluorescence microscopy (**Fig 8F and 8G**). Moreover, TM-UBE2G2 co-immunoprecipitated both HRD1 and SEL-1L, as previously shown for AUP1 (**S6C Fig**) [35,39]. Importantly, membrane-tethered UBE2G2 is functional in vivo as it reconstituted ERAD of NHK in UBE2G2 KO cells (**S6D Fig**). By contrast, when TM-UBE2G2 was expressed in AUP1 KO cells it did not, by itself, restore ERAD (**Fig 8H, lanes 4 to 6**). However, degradation of NHK was fully restored when TM-UBE2G2 was co-expressed with G2BR$^{AUP1}$ (GFP-G2BR$^{AUP1}$) (**Fig 8H, lanes 7 to 9**; for a direct comparison of the level of TM-UBE2G2 to endogenous UBE2G2, see **S6B Fig, lanes 1, 4, and 5**). Similarly, TM-UBE2G2 together with GFP-G2BR$^{AUP1}$ restored degradation of RI$_{332}$ (**Fig 8I, lanes 7 to 9**). This requirement for the G2BR builds upon the in vitro biophysical and biochemical findings in this and previous studies and demonstrates that the G2BR indeed has an activating role in cells. Taken together, our results suggest three distinct cellular roles for G2BR$^{AUP1}$: (1) stabilization of UBE2G2 by preventing its apparent self-targeting for degradation; (2) recruitment of UBE2G2 to the ER membrane; and (3) activation of UBE2G2 as an ERAD E2.

## Discussion

In this study, we have evaluated the domain requirements and roles of AUP1 in the degradation of two well-studied HRD1 ERAD-L substrates, NHK and RI$_{332}$. We also assessed the ERAD-C-dependent substrate GFP$^u$, the degradation of which has been reported in fluorescence screens to involve TRC8 as well as MARCH6 [40,41]. In addition, we evaluated the requirement for AUP1 in the basal degradation of a presumed ERAD-M substrate, HMGCR. For none of these substrates is the acyltransferase or CUE domain of AUP1 necessary for degradation. The hairpin membrane anchor of AUP1 is similarly dispensable if an alternative means of recruiting UBE2G2 to the ER membrane is provided. In striking contrast, the carboxyl-terminal, 27-aa UBE2G2 Binding Region (G2BR$^{AUP1}$) of AUP1 is absolutely required for degradation of these substrates. This is similar to our observations in yeast, where the U7BR is the only domain of Cue1p required for ERAD [18]. Consistent with a previous report [40], we found AUP1 to be completely dispensable for the degradation of INSIG-1, which is an ERAD-M substrate that is targeted by gp78. Importantly, gp78 has its own G2BR that we now show is required for degradation of this substrate. Thus, the G2BR represents a conserved domain that is implicated in all three characterized ERAD pathways.

The crystal structure of the UBE2G2:G2BR$^{AUP1}$ described herein closely resembles the previously reported UBE2G2:G2BR$^{gp78}$ structure [17,30] and reveals that G2BR residues which contact UBE2G2 are highly conserved. Additionally, the affinities of G2BR$^{AUP1}$ and G2BR$^{gp78}$ for UBE2G2 are similar. The low nanomolar affinity of G2BR$^{AUP1}$ for UBE2G2 is driven by a combination of salt bridges, hydrogen bonds, and hydrophobic interactions distributed along the length of the UBE2G2:G2BR$^{AUP1}$ interface as revealed by the crystal structure. Moreover, the allosteric enhancement of RING domain binding by UBE2G2 is equivalently induced by G2BR$^{AUP1}$ compared to G2BR$^{gp78}$. We also find that G2BR$^{AUP1}$ binds selectively to UBE2G2 and mutations in G2BR$^{AUP1}$ that lead to a loss of UBE2G2 binding in vitro correlate with a loss of substrate degradation in cells. These findings are significant, given studies implicating the transmembrane ERAD E2, UBE2J1, in functioning together with AUP1 in the processing of ERAD-L and ERAD-C substrates [35,40,54,64]. Interestingly, the *S. cerevisiae* Cue1p U7BR has a more complex structure than the mammalian G2BRs, which includes three helices that are essential for binding Ubc7p [19]. By contrast, the recently described structure of the *Schizosaccharomyces pombe* U7BR bound to its cognate E2 closely resembles the single α-helix of G2BR$^{AUP1}$ and G2BR$^{gp78}$ [71].

We have uncovered three distinct functions for G2BR^AUP1 with respect to UBE2G2, the first of which is a role in maintaining UBE2G2 levels. Loss of AUP1 expression results in decreased levels and rapid degradation of catalytically active UBE2G2, consistent with a recent observation on steady state UBE2G2 levels [41]. However, restoring UBE2G2 levels by its over-expression is not sufficient to reconstitute ERAD. Reexpressing either the G2BR^AUP1 or G2BR^gp78 as GFP fusions protects UBE2G2 from degradation but similarly does not restore ERAD. This is in contrast to yeast lacking Cue1p, where soluble Cue1p fragments containing an N-terminally extended U7BR both stabilize Ubc7p and reconstitute ERAD [18,70]. Interestingly, unlike loss of AUP1, loss of gp78 does not result in rapid degradation of UBE2G2. Although the mechanism responsible for the stabilizing effect of G2BRs on UBE2G2 requires further study, the differential effects of loss of AUP1 and gp78 may be, at least in part, a conse-quence of the relative abundance of the two proteins in the cell. Our analysis suggests that there is nearly 40-fold more AUP1 than gp78 in HT1080 cells. A decrease in UBE2G2 levels cannot fully account for the requirement for AUP1 in ERAD. However, it is reasonable to con-sider that AUP1 may have a context-dependent function as a "rheostat," determining UBE2G2 levels and thereby the capacity of cells to efficiently carry out ERAD.

A second function of the G2BR^AUP1 is to bring UBE2G2 to the ER membrane. Expression of AUP1 lacking both the acyltransferase and CUE domains, essentially a membrane-anchored G2BR, in AUP1 KO cells is sufficient for the degradation of all ERAD substrates assessed. By contrast, we find that expression of soluble GFP-G2BR fusions or AUP1 lacking only its mem-brane anchor does not reconstitute ERAD. This represents a dichotomy with findings in yeast (see preceding paragraph) [18]. We also know, indirectly, that there is apparently nothing unique about the AUP1 hairpin domain in the context of ERAD, as degradation of NHK can be reconstituted by tethering UBE2G2 to the ER membrane using the Cue1p transmembrane domain, as long as the G2BR^AUP1 is provided in *trans*.

The ability to reconstitute ERAD in cells lacking AUP1 when a G2BR is provided in a solu-ble form and UBE2G2 is tethered to the membrane reveals a third function of the G2BR, which is to activate UBE2G2. Our extensive analysis of the G2BR^gp78 [17], as well as data pre-sented here for G2BR^AUP1, demonstrates that the two G2BRs allosterically enhance the affinity of E2:RING interactions, which, in turn, dramatically enhances ubiquitination in vitro. While we cannot directly assess the exact activating role in cells, the fact that expression of UBE2G2 in a membrane-anchored form is insufficient to restore ERAD, yet this function can be recon-stituted simply by co-expressing a G2BR without any additional AUP1 sequence, provides strong corroborating evidence for an activating effect.

Prior to this study, there has not been, as far as we are aware, any direct assessment of the role of AUP1 on substrate ubiquitination. We show that loss of AUP1 or of UBE2G2 dramati-cally decreases substrate ubiquitination. Thus, whether or not AUP1 is directly involved in ret-rotranslocation of substrates, as proposed [54,72], it clearly contributes to substrate ubiquitination. Ubiquitin ligases are well known to either undergo autoubiquitination or be subject to ubiquitination by other E3s [73]. Studies in yeast in which Ubc7p and the U7BR domain of Cue1p are used to auto-ubiquitinate Hrd1p in vitro have provided evidence that ubiquitination of this E3 contributes to its proposed role as a channel for retrotranslocation of ERAD-L substrates [66–68,74]. We observe robust constitutive polyubiquitination of endoge-nous human HRD1. Interestingly, however, this ubiquitination is not altered in intensity or pattern by loss of either AUP1 or UBE2G2. It is therefore evident that other E2s are primarily responsible for HRD1 ubiquitination in vivo. One such candidate is UBE2J1, which has been implicated in functioning together with components of the HRD1 ERAD machinery [39,54,64,75,76]. It may be that these two E2s are playing roles in different, albeit coupled, aspects of HRD1-mediated ERAD. In this regard, although loss of AUP1 or UBE2G2 markedly

decreases substrate ubiquitination, we do not exclude roles for other cooperating or initiating E2s, including UBE2J1, in this process. Moreover, the striking level of HRD1 ubiquitination that we observe may provide a molecular explanation for the reported requirement for the CUE domain of AUP1 in physical interactions with the HRD1 degradative machinery [35]. It will be of interest going forward to further characterize this HRD1 ubiquitination and determine whether it is playing a role in retrotranslocation of ERAD-L substrates.

We find neither the AUP1 CUE nor the acyltransferase domain to be required for substrate degradation. This seeming indifference to the AUP1 CUE domain stands in contrast to gp78 where the CUE domain is required for substrate degradation [23,28]. However, we cannot exclude subtle kinetic effects of the AUP1 CUE domain on degradation that might be revealed through assessment of different substrates. Additionally, there may be effects on ubiquitin chain elongation on substrates as reported for the CUE domain of Cue1p [15,16,77]. Our understanding of the roles of the ubiquitination machinery in ERAD, particularly in mammalian cells, is still very much a work in progress. Despite all of the studies that have been carried out, most of the substrates evaluated are either artificial test substrates or mutations of cellular proteins that are being exploited to understand the details of the process. Thus, it is important going forward that neither the acyltransferase nor the CUE domain be excluded from consideration in assessing proteostasis in the secretory pathway. However, what is conclusively established from our work is that the G2BR plays overt and essential roles in this process.

## Materials and methods

### Cell lines

HT1080 (ATCC, CCL-121), HEK293, U-2 OS (ATCC, HTB-96), 293T (ATCC, CRL-3216), and HeLa (CCL-2) were maintained in DMEM supplemented with 10% (v/v) fetal bovine serum, 100 units/ml penicillin, 100 μg/ml streptomycin, and 2 mM glutamine in a humidified incubator at 37°C and 5% $CO_2$. BE(2)- M17 (ATCC, CRL2267) cells were additionally supplemented with MEM nonessential aa (HyClone), 100 μM sodium pyruvate, and 50 μM 2-mercaptoethanol (Gibco, Waltham, MA, USA). CRISPR/Cas-9 KO variants of HT1080, M17, and HEK293 were generated using Santa Cruz Double Nickase Plasmid kits: AUP1 (sc-410699-NIC), gp78 (sc-402346-NIC), and UBE2G2 (sc-404851-NIC), Ultra Cruz transfection reagent and plasmid medium (Santa Cruz, Dallas, TX, USA). Transfected cells were selected with 2 μg/mL puromycin (Millipore Sigma, Burlington, MA, USA) for approximately 3 weeks. Individual clones were screened, and loss of protein expression was confirmed by western blotting. sgRNA targeting HRD1 early in the protein sequence were identified using the sgRNA Scorer 2.0 algorithm [78] and 6 candidates were in vitro transcribed, complexed with Cas-9 protein and evaluated in 293T cells for editing efficiency as previous described [79]. The best candidate was then cloned into the LentiCRISPRv2GFP vector and transfected into parental cells using Lipofectamine 2000 (Thermo Fisher, Waltham, MA, USA). LentiCRISPRv2GFP was a gift from David Feldser (Addgene plasmid #82416; http://n2t.net/addgene:82416; RRID:Addgene_82416) [80]. Transfected cells were sorted by fluorescence activated cell sorting, and individual clones were expanded for sequencing to confirm HRD1 gene disruption. PCR primers (forward: 5′-TCCCTACACG ACGCTCTTCCGATCT**ACACCAGTTCTACCCCACTG**-3′ and reverse: 5′-GTTCAGA CGTGTGCTCTTCCGATCT**TGAAGAGTGCAACAAAGCGG**-3′, where bold sequence corresponds to locus-specific sequence) were used to amplify around the region of interest and amplicons were subjected to Illumina sequencing. Loss of HRD1 protein expression was also confirmed by western blotting.

## Plasmid constructs

Sequences encoding aa 61 to 410, 292 to 410, and 379 to 410 of AUP1 were amplified by polymerase chain reaction (PCR) from pEGFP-AUP1 (a generous gift from Hidde Ploegh) [54] and subcloned into pGEX-6P-1 (GE Healthcare, Chicago, IL, USA) from BamHI to XhoI. For co-expression of G2BR$^{AUP1}$ and UBE2G2 in *Escherichia coli*, sequence encoding aa 379 to 410 of AUP1 was subcloned from BamHI to HindIII sites within the first multiple cloning site (MCS) of pETDuet-GST [81]. UBE2G2 was subcloned from NdeI to XhoI within the second MCS of the vector. Sequence encoding full-length AUP1 was amplified by PCR from pEGFP-AUP1 and subcloned into pcDNA3-FLAG from BamHI to XhoI. All mutations in pGEX-6P-1 and pcDNA3-AUP1 were generated by site-directed mutagenesis using Quikchange XL (Agilent Technologies, Santa Clara, CA, USA), including 2A (R382A/Q383A), 2E (R382E/Q383E), 5E (R382E/Q383E/K390E/R398E/R400E), L386D, Y394D, L386D/Y394D, and A397D. Deletions of the transmembrane region (ΔTM, aa 25 to 45), G2BR (ΔG2BR, aa 386 to 397), CUE domain (ΔCUE, aa 293 to 337), and both the acyltransferase and CUE domains (ΔATΔCUE, 90 to 337) were generated in pcDNA3-AUP1-FLAG using QuikChange XL. Nucleotide sequences encoding AUP1 (aa 378 to 404), gp78 (aa 574 to 600) or a scrambled (SCR) gp78 sequence (SKANDSRELQFRKMRLRAQRQVQELDKL) were synthesized (Integrated DNA Technologies, Coralville, IO, USA) and subcloned into pEGFP-C1 vector (Clontech, San Jose, CA, USA) from BamHI to XhoI to generate GFP-tagged G2BRs. Sequence encoding the cytoplasmic domain of HRD1/SYVN (aa 236 to 617) was subcloned from pcDNA3.1(+)-HRD1-Myc/6His A(−) [31] into pGEX-4T-3 (GE Healthcare) from BamHI to EcoRI. Sequence encoding HRD1 (aa 236 to 444) was subsequently PCR amplified and subcloned into pGEX-6P-1 from BamHI to EcoRI. The gp78-RING construct (aa 313 to 393) was generated by inserting a stop codon in the pGEX-4T-2-gp78C plasmid [25]. gp78 (aa 429 to 611) was PCR amplified from pGEX-4T-2-gp78C and subcloned into the pGEX-6P-1 from BamHI to SalI. Site-directed mutagenesis of sequence corresponding to aa 595 to 600 within the G2BR (KRFLNK to AGAAGG) in pCINeo-gp78 [25] was used to generate a plasmid encoding gp78 with a disrupted G2BR. Disruption of this region in G2BR$^{gp78}$ results in a $>5 \times 10^3$ decrease in its affinity for UBE2G2 [17]. Sequence encoding the cytoplasmic domain of TRC8 (aa 513 to 664) was PCR amplified from HA-TRC8 [82] and subcloned into pGEX-4T-3 from BamHI to EcoRI. pcDNA3-NHK-HA was provided by Jeff Brodsky and was a gift from Ron Kopito [39]. NHK was PCR amplified and cloned into pcDNA4/MYC-6His A from HindIII to XhoI to generate NHK-MYC-6His. Ribophorin I (aa 1 to 332) was PCR amplified from pCMV-SPORT6-RPN1 (Dharmacon, Lafayette, CO, USA) and cloned into pcDNA4/MYC-6His A from EcoRI to XbaI. pcDNA3.1(+)-Cue1pTM-MYC-UBE2G2 (TM-UBE2G2) was generated by PCR of sequence encoding aa 1 to 44 of Cue1p from pRS316-Cue1p-Ubc7-HA [18] and subcloning into pcDNA3.1(+)-MYC-UBE2G2 from BamHI to EcoRI. This construct was modified by site-directed mutagenesis to add a carboxyl-terminal FLAG tag for efficient immunoprecipitation, generating pcDNA3.1(+)-Cue1pTM-MYC-UBE2G2-FLAG (TM-UBE2G2-FLAG). Sequence encoding the cytoplasmic domain of UBE2J1 (aa 1 to 283) was PCR amplified from an EST (IMAGE #4137664) and cloned into pET3a (Novagen, Burlington, MA, USA) from NdeI to BamHI. A Cys residue was inserted between the PreScission Protease cleavage sequence and HA-tag (LEVLFQGPLGS**C**YPYDVPDYA) on pGEX-6P-1-HA-ubiquitin [81] by site directed mutagenesis to allow labeling with Cy5. The deduced aa sequence of all coding regions was confirmed by sequencing.

The following plasmids have been previously described: pGEX-4T-2-UBE2G2, pcDNA-HA-UBE2J2, pcDNA-MYC-UBE2G2 WT and C89S [29]; pET3a-UBE2G2 [17]; pcDNA3.1(+)-HA-ubiquitin [83]; pET15b-UBE2D2 and pET15b-UBE2D3 [65]; pCIneo-gp78 [25]; pCMV14-INSIG-1-MYC was a gift from Joseph Goldstein [84]; pET11d-UBE2G1 was a gift

from Simon Wing [85]; GFP$^u$ was a gift from Ron Kopito [45]; and LentiCRISPRv2GFP (Addgene #82416) was a gift from David Feldser [80].

## Antibodies

Antibodies were obtained for HRD1/SYVN1 (D3O2A #14773), AUP1 (D5M9Q #35055), UBE2G2 (D8Z4G #63182), CHOP (L63F7, #2895), BiP (C50B12, #3177), Phospho-eIF2α (S51, D9G0, #3398), eIF2α (#9722), and β-actin (8H10D10, #3700) from Cell Signaling (Danvers, MA, USA); FLAG (M2, F1084) and HA-Peroxidase (clone 3F10) from Millipore Sigma; GFP (B-2, sc-9996), Calnexin (AF18, sc-23954), HA (Y-11, sc-805), Sel-1L (F-3, sc377350) and ubiquitin (P4D1, sc-8017) from Santa Cruz; HSP60 (ab46798) from Abcam (Waltham, MA, USA); Luciferase (G7451) from Promega (Madison, WI, USA); HRP-linked anti-Mouse (NA931) and anti-Rabbit (NA934) IgG from GE Healthcare; Fluorescein-conjugated anti-Mouse IgG (F2761) and Texas Red-X-conjugated anti-Rabbit IgG (T6391) from Invitrogen (Waltham, MA, USA); and Clean-Blot (HRP) from Thermo Fisher. Rabbit antibodies recognizing the G2BR$^{gp78}$ (Ab2) was raised to aa 574 to 611 of gp78 and affinity purified using a peptide corresponding to aa 574 to 597 of gp78. In **S1A Fig**, where the G2BR$^{gp78}$ is disrupted, affinity-purified rabbit Ab1 antibodies, raised against aa 505 to 529 of gp78, were utilized [25]. Rabbit antiserum against HRD1 [23], antibodies against gp78 (Ab2) [28], and ubiquitin [86] were previously described. Anti-MYC (9E10, ATCC CRL-1729) was purified on a MYC column from hybridoma supernatant. Preparation and use of mouse monoclonal antibody directed against HMGCR (A9) has been described [55]. Anti-HA Affinity Gel (E6779), Anti-c-MYC Affinity Gel (A7470), and Anti-FLAG M2 Magnetic Beads (M8823) were from Millipore Sigma.

## Peptides

G2BR peptides (**Table 2**) were synthesized as follows for in vitro and biophysical measurements.

G2BR$^{gp78}$ was synthesized with the addition of an N-terminal Trp residue for peptide quantification. Scrambled peptide (SCR) is a mutated sequence based on aa 378 to 404 of AUP1 and used as a negative control in experiments. Lyophilized peptides were resuspended and dialyzed extensively against 50 mM Tris-HCl, pH 7.4 at 4˚C, and their concentrations determined by absorbance at 280 nm using calculated extinction coefficients from aa sequences. Dialyzed peptides were stored in small aliquots at −20˚C.

## Cellular experiments

Plasmid transfections were carried out using PolyFect (Qiagen, Germantown, MD, USA) or JetPrime (Polyplus, New York, NY, USA) reagents. For CHX chase experiments, approximately 24 hours following plasmid transfection, cells were treated with 50 μg/mL CHX in complete media for the indicated times. For lipid droplet experiments, cells were treated with 200 μM oleic acid (Millipore Sigma) for 16 hours, beginning 24 hours following transfection.

**Table 2. Amino acid sequences of G2BR peptides.**

| G2BR | aa residues | Sequence | Source |
|---|---|---|---|
| gp78 | 574 to 600 | WSADERQRMLVQRKDELLQQARKRFLNK | GenScript |
| AUP1 | 378 to 404 | SSWARQESLQERKQALYEYARRRFTER | GenScript |
| SCR | 378 to 404 | SSWAYYAYKFASERERAASKQQSLTER | PEPTIDE 2.0 |

aa, amino acid; AUP1, ancient ubiquitous protein 1; G2BR, UBE2G2 Binding Region.

For assessment of GFP$^u$ stability, cells were cultured in DMEM supplemented with 10% Fetal-Plex animal serum complex (Gemini Bio, West Sacramento, CA, USA) and treated with 50 μM emetine for the indicated times to inhibit protein synthesis [40]. For experiments assessing basal degradation of HMGCR, expression of HMGCR was induced by culturing cells for 24 hours in LPDS media [55], which contains DMEM supplemented with 10% (v/v) fetal bovine lipoprotein-deficient serum (LPDS; $d > 1.25$), 100 μM sodium mevalonate, and 5 μM compactin. Cells were then treated with 50 μg/mL CHX in LPDS media for the indicated duration and cell lysates were harvested. LPDS was prepared by ultracentrifugation, as described [87]. For assessment of whole cell lysates, cells were lysed in modified RIPA buffer (1X PBS pH 7.4, 1% Triton X-100, 0.5% sodium deoxycholate) supplemented with 25 μM MG132 and cOmplete Protease Inhibitor Cocktail (Roche, Burlington, MA, USA). For co-immunoprecipitation with Cue1pTM-UBE2G2-FLAG, cells were treated with 25 μM MG132 for 4 hours, lysed in PBS (pH 7.4) with 1% digitonin and 40 μM MG132, and lysates immunoprecipitated with Anti-FLAG (M2) Magnetic Beads. For co-immunoprecipitation with FLAG-UBE2G2, cells were lysed in PBS (pH 7.4) with 0.5% Triton X-100, 0.1% deoxycholate and 40 μM MG132, and immunoprecipitated with Anti-FLAG (M2) magnetic beads. Clarified lysates or immunoprecipitates were denatured by heating in NuPAGE LDS Sample Buffer with β-mercaptoethanol at 70°C for 10 minutes, resolved on Bis-Tris polyacrylamide gels in NuPAGE MES or MOPS SDS running buffer and analyzed by western blotting. Proteins were detected using Clarity Western ECL (Bio-Rad, Hercules, CA, USA), Radiance Q (Azure Biosystems, Dublin, CA, USA), or SuperSignal Femto West (Thermo Scientific, Waltham, MA, USA) ECL reagents and a c280 Imager (Azure Biosystems) or autoradiography.

**Cellular ubiquitination assays.** Approximately 24 hours after transfection with NHK-MYC-6His and HA-ubiquitin plasmids, cells were treated with 25 μM MG132 for 4 hours and lysed in denaturing buffer (50 mM Tris-HCl pH 7.4, 8 M urea, 1% Triton X-100) supplemented with 10 mM iodoacetamide (Millipore Sigma), 25 μM MG132. Equivalent amounts of NHK-MYC-6His were enriched with Ni-NTA beads (Qiagen), washed with denaturing buffer containing 20 mM imidazole, quenched in LDS sample buffer, and analyzed by western blot. For HRD1 ubiquitination, cells were transfected with vector or HRD1-MYC-6His and HA-ubiquitin, lysed, and enriched as described above. Supernatant from the first pull-down was subjected to second pull-down with Ni beads to confirm the efficiency of the first, and samples were analyzed by western blot. For ubiquitination of endogenous HRD1, cells were lysed in 1% SDS in 50 mM Tris-HCl (pH 7.4) supplemented with 10 mM iodoacetamide and diluted with modified RIPA buffer (50 mM Tris-HCl pH 7.4, 150 mM NaCl, 1% Triton X-100, 0.1% sodium deoxycholate) containing 10 mM iodoacetamide and 40 μM MG132. Equivalent amounts of HRD1 were immunoprecipitated with HRD1 antiserum.

**In vivo quantitation of AUP1 and gp78.** GST-AUP1 (aa 61 to 410) and GST-gp78 (aa 429 to 611) were expressed in *E. coli* BL21-DE3 cells for 16 hours at 20°C. Cell pellets were resuspended in 50 mM Tris-HCl, 1% Triton X-100, 0.5 mM EDTA, 5 mM DTT, and protease inhibitors (Roche) and lysed by sonication. GST fusion proteins were purified with Glutathione Sepharose (GS) 4B and quantified by SDS-PAGE and Coomassie staining. To generate a standard curve for the AUP1 or gp78 antibody, known quantities of purified GST-AUP1 (1 to 20 ng) or GST-gp78 (0.05 to 0.5 ng) were resolved alongside $1.1 \times 10^4$ cellular equivalents of HT1080 lysate by SDS-PAGE. Cellular protein levels were estimated based on their immunoreactivity to gp78 and AUP1 antibodies and quantification of band density relative to the known GST fusion standards. Cellular protein levels were converted from ng to moles, adjusting for differences in molecular weight between the endogenous and GST fusion, so that estimated concentrations of gp78 and AUP1 could be compared directly.

## Membrane fractionation

Membrane fractionation experiments were previously described [23,55]. Twenty-four hours after transfection, cells were washed and resuspended in 0.25 M sucrose, 10 mM triethanolamine, supplemented with 25 μM MG132 and protease inhibitors, and then passed through a 27.5-gauge needle 15 times. Cell homogenates were centrifuged twice for 5 minutes at $1,000 \times g$ to remove the nuclei, and the supernatants were spun twice for 30 minutes at $100,000 \times g$ at 4˚C in a Sorvall Discovery M150 ultracentrifuge using a S100 AT4-541 rotor to separate membrane (pellet) and cytosolic (supernatant) components. Cytosolic and membrane fractions were resolved on Bis-Tris polyacrylamide gels and analyzed by western blot.

## Fluorescence microscopy

**Subcellular localization.** For subcellular localization studies, U2-OS or HT1080 AUP1 KO cells cultured on glass coverslip in 6-well plates were transfected with plasmid encoding pcDNA-MYC-TM-UBE2G2 and HA-UBE2J2. After 24 hours, cells were fixed for 10 minutes in 4% paraformaldehyde preheated to 37˚C, washed in PBS (pH 7.4), and permeabilized with 0.2% Triton X-100 in PBS for 5 minutes at room temperature. Nonspecific adsorption was reduced by blocking in PBST (PBS pH 7.4 containing 0.05% Tween-20) with 10% BSA for 1 hour. MYC-TM-UBE2G2 was detected by incubating with mouse anti-MYC (9E10) and HA-UBE2J2 with rabbit anti-HA (Y-11) antibody in PBST with 2% BSA overnight at 4˚C. Coverslips were washed with PBST and incubated for 1 hour at room temperature with Fluorescein-conjugated anti-Mouse IgG (1:1,000; F2761) and Texas Red-X-conjugated anti-Rabbit IgG (1:1,000; T6391) antibodies in PBST. After washing 3 times with PBST, the coverslips were mounted onto a glass slide in mounting medium (90% glycerol in 1M Tris-HCl pH 8.5, 1 mg/mL p-Phenylenediamine) supplemented with 100 nm TetraSpeck beads (Invitrogen, T7279) at 1:600 dilution.

**Lipid droplets.** For staining of lipid droplets, HT1080 cells were plated in 35-mm glass bottom dishes (ibidi, μ-Dish 35-mm high Glass Bottom) and treated with 200 μM oleic acid overnight. Cells were then fixed for 10 minutes in 4% paraformaldehyde preheated to 37˚C, washed in PBS (pH 7.4) and permeabilized with 0.2% saponin (Nacalai USA, San Diego, CA, USA) in PBS for 30 minutes at room temperature. Lipid droplets, F-actin, and nuclei were stained with LipidSpot 488 (Biotium, Fremont, CA, USA), Phalloidin-ATTO643 (ATTO-TEC, Siegen, Germany), and Hoechst 33342 (Thermo Fisher), respectively.

**Confocal microscopy.** Confocal images were acquired on a Leica DMi8 microscope equipped with Yokogawa CSU-W1 Spinning Disk Confocal and Andor Zyla 4.2 sCMOS camera controlled with Andor Fusion software. Images were acquired in 0.1 μm sections (for deconvolution) or 0.2 μm otherwise with a 100× objective (NA 1.4). When indicated, images were deconvolved in Fusion software on default settings. For co-localization, signals from TetraSpeck beads included in the mounting media were used as positive control.

**SIM.** Structured illumination microscopy (SIM) images were acquired on N-SIM (Nikon, Minato City, Tokyo, Japan) using an Apo TIRF 100X (NA 1.49) Plan Apo oil objective. Image stacks were acquired in 3D-SIM mode with a z-distance of 0.1 μm, and the raw images (15 per plane: 5 phases, 3 angles) were then reconstructed to generate a super-resolution image. All images were processed the same way. Signals from TetraSpeck beads included in the mounting media were used for color registration after image reconstruction.

## qPCR

RNA was extracted from HT1080 cells with TRIzol (Invitrogen) according to manufacturer's instructions and reverse transcribed to cDNA using SuperScript III First-Strand Synthesis

System for RT-PCR (Invitrogen). PCR reactions were performed on a CFX96 Real-Time PCR system using iQ SYBR Green Supermix (Bio-Rad). Transfected AUP1-FLAG constructs were amplified using the forward and reverse primers 5′-CCTGAAGACATCACCAAGGGA-3′ and 5′-GTGGTGCTCGAAGATCTTGTC-3′, respectively. UBE2G2 and HRD1/SYVN were amplified using RT$^2$ quantitative polymerase chain reaction (qPCR) Primer Assay (Qiagen) using NM_182668.2 and NM_172230, respectively. ER stress markers were amplified using the following primers (IDT): ATF-4: 5′-ATGGCCGGCTATGGATGAT-3′ and 5′-CGAAGTCAAAC TCTTTCAGATCCATT-3′; BiP: 5′-TGTTCAACCAATTATCAGCAAACTCTTCTGCTGTA TCCTCTTCACCAGT-3′ and 5′-ACTGGTGAAGAGGATACAGCAGAAGAGTTTGCTGA TAATTGGTTGAACA-3′; CHOP: 5′-CTGCCTTTCACCTTGGAGAC-3′ and 5′-CGTTTCC TGGGGATGAGATA-3′; and XBP1-spliced: 5′-GAGTCCGCAGCAGGTG-3′ and 5′-GTGTC AGAGTCCATGGGA-3′. A total of 3 technical replicates for each condition were measured and analyzed by the ΔΔCt method using either β-actin or GAPDH as an internal control (QuantiTect primers QT01680476).

### In vitro E2 binding assay

In vitro translated $^{35}$S methionine-labeled E2s were generated using the *E. coli* T7-S30 Extract System (Promega) and excess $^{35}$S methionine was removed by desalting twice with Zeba Spin Desalting Columns (Thermo Scientific). GST fusion proteins were quantified by Coomassie blue staining. Moreover, 0.1 μg of GST fusion protein (approximately 3 pmol) was prebound to GS 4B (GE Healthcare) and then incubated with approximately $10^5$ cpm of $^{35}$S methionine-labeled E2 (approximately 0.2 pmol) in 200 μl Binding Buffer (25 mM Tris-HCl, pH 7.4, 50 mM NaCl, 5 mM DTT, 0.5% IGEPAL CA-630) overnight at 4˚C. Beads were extensively washed in Binding Buffer, resolved on Bis/Tris polyacrylamide gels under reducing conditions, dried and visualized on a Typhoon FLA 7000 (GE Healthcare). Binding was quantified using ImageQuant Software.

### In vitro ubiquitination assay

GST-Cys-HA-ubiquitin was expressed in *E. coli* BL21 (DE3), purified with GS 4B, and incubated with excess Cy5 maleimide (Amersham PA15131) at room temperature for 1 hour. After removing excess dye by extensive washing, Cy5-HA-ubiquitin was released from the beads with PreScission Protease (GE Healthcare) overnight at 4˚C, leaving the N-terminal sequence (GPLGS) prior to the labeled Cys. Cy5-labeled ubiquitin was quantified with Coomassie staining. Ubiquitination reactions were performed with 1 μM of E3 immobilized on GS 4B in 50 μl reaction volume. The reaction mixture contained approximately 100 nM E1, 2 μM UBE2G2, 8 μM Cy5-labeled ubiquitin, and 5 μM peptide in reaction buffer (50 mM Tris-HCl pH 7.4, 5 mM MgCl$_2$, 20 mM ATP, 1 mM DTT, and 0.1% Triton X-100). Reactions were carried out at 37˚C for 90 minutes with continuous shaking, quenched with 4X NuPAGE LDS Sample Buffer (Thermo Fisher), resolved on Bis/Tris polyacrylamide gels under reducing conditions, and visualized on a Typhoon FLA 7000.

### In vitro deubiquitination

HT1080 cells were lysed in lysis buffer (50 mM Tris-HCl pH 7.4, 120 mM NaCl, 1% Triton X-100) supplemented with 10 mM iodoacetamide, protease inhibitors, and 40 μM MG132. Lysate was clarified by centrifugation at 1,000 x g at 4˚C for 10 minutes. Ubiquitin conjugates were enriched by binding with TUBE1 and TUBE2 agarose (UM401, UM402, respectively; Life Sensors, Malvern, PA, USA) on a rotating platform for 4 hours at 4˚C. Control agarose (UM400) was used as negative control. After washing 3× with lysis buffer (20× bed volume), the beads

were treated with 5 mM DTT in 0.1% Triton X-100 buffer (50 mM Tris-HCl pH 7.4, 100 mM NaCl, 0.1% Triton X-100) for 15 minutes and washed 2x with buffer containing 1 mM DTT. The beads were then treated with buffer alone or a DUB cocktail (DB599, Life Sensors) at 37°C for 4 hours with constant agitation. The beads were washed 3× with 0.1% Triton X-100 buffer and bound proteins were eluted with 2X sample buffer.

## Protein expression for crystallization

The pETDuet-GST vector was used to co-express GST-AUP1(379 to 410) and UBE2G2 in *E. coli* BL21(DE3) cells (Invitrogen). Briefly, cells were induced at $OD_{600}$ 0.8 with 0.15 mM IPTG overnight at 25°C to express both proteins. Cells were lysed with an APV-2000 homogenizer in binding buffer (50 mM Tris-HCl pH 7.5, 150 mM NaCl, 1 mM DTT), and the homogenate was centrifuged at 27,000 x g for 20 minutes to remove bacterial cell wall. The supernatant was collected and incubated with GS 4B for 2 hours at 4°C to isolate GST-AUP1(379 to 410) and the associated UBE2G2. The beads were then collected by centrifugation and extensively washed with binding buffer. The GST tag was removed through on-column cleavage with PreScission Protease overnight at 4°C. After purification, the complex containing AUP1(379 to 410) and UBE2G2 was immediately used for crystallization.

## Crystallization, data collection, and structure determination

Crystals of the UBE2G2:G2BR$^{AUP1}$ complex were grown by sitting drop vapor diffusion at 19 ± 1°C. Each droplet contained 0.6-μl protein solution (15 mg/ml in 150 mM NaCl and 50 mM Tris-HCl, pH 7.5) and 0.2-μl reservoir solution (30% PEG 4000, 0.2 M NH$_4$Ac, and 0.1 M NaAc, pH 4.6), and the premixed 0.8 μl droplets were equilibrated against 60 μl reservoir solution. The rod-shaped crystals grew to full size (0.08 mm × 0.08 mm × 0.3 mm) in 3 days. Crystals were flash-cooled in cold nitrogen stream. X-ray diffraction data were collected from a single crystal on a MARMOSAIC 325 detector at the BL9-2 beamline of the Stanford Synchrotron Radiation Lightsource (SSRL) and processed using the HKL-3000 program suite [88]. Data statistics are presented in **Table 1**. The UBE2G2:G2BR$^{AUP1}$ structure was determined by molecular replacement using phenix.automr of the PHENIX program suite [89]. The crystal structure of the UBE2G2:G2BR$^{gp78}$ (PDB entry 3H8K) [17] was used as the search model after solvent molecules were removed. The structure was refined with phenix.refine of PHENIX, and the model building and adjustment was done with COOT [90]. About 1,000 reflections were randomly selected for cross-validation. Water molecules were included at the last stage of the refinement on the basis of difference electron density (Fo-Fc, above 3σ) and verified with omit maps. The refined structure was validated using the PROCHECK [91] and WHATIF [92] programs. Figures were generated with PyMol (DeLano Scientific, South San Francisco, CA, USA). Refinement statistics of the crystal structure are also presented in **Table 1**.

## NMR

UBE2G2 and gp78 RING used in titration experiments were expressed and purified as previously described [24]. G2BR$^{gp78}$ and G2BR$^{AUP1}$ peptides were solubilized and dialyzed against 50 mM Tris-Cl pH 7.0, 0.5 mM TCEP, 0.05% sodium azide. NMR samples of G2BR$^{gp78}$:UBE2G2 and G2BR$^{AUP1}$:UBE2G2 were prepared in 50 mM Tris-Cl pH 7.0, 0.5 mM TCEP, 100 μM zinc sulfate, and 0.05% sodium azide. NMR spectra were collected on Bruker AVIII spectrometers operating at 800 or 850 MHz using cryogenic TCI probes. The $^1$H-$^{15}$N HSQC spectra of $^{15}$N-labeled UBE2G2 at 150 μM were collected in the absence and presence of gp78 RING at concentrations ranging between 0 and 450 μM. The changes in chemical shift for each peak in the 2D spectrum upon gp78 RING binding was calculated using the following

CSP equation:

$$\Delta CSP = \sqrt{\frac{\left(\frac{\Delta\delta_N}{5}\right)^2 + (\Delta\delta_H)^2}{2}} \tag{1}$$

The CSPs calculated from Eq 1 were plotted versus the molar ratio of gp78 RING to UBE2G2:G2BR, and the data were fit using the following equation to calculate the dissociation constant ($K_d$), where x is the ligand-to-protein ratio:

$$\Delta CSP = \frac{\Delta CSP_{max}}{2} * \sqrt{1 + x + \frac{K_d}{P} - \frac{(1 + x + K_d)^2}{P} - 4x} \tag{2}$$

## ITC

ITC measurements were performed with purified UBE2G2 [17] and G2BR$^{gp78}$ or G2BR$^{AUP1}$ peptides (GenScript, Piscataway, NJ, USA). Samples were extensively dialyzed against PBS (pH 7.4) at 4°C, and final concentrations were determined by absorbance at 280 nm. UBE2G2: G2BR interactions were studied using an iTC200 calorimeter (Malvern/MicroCal) at 25°C. A typical experiment included injection of 18 aliquots (2.1 μL each) of 30 μM G2BR into a solution of 3 μM UBE2G2 in the cell (volume 200 μL) at a stirring speed of 750 RPM. An additional set of injections was run in a separate experiment with buffer in the cell instead of the protein solution. These blank experiment data were subtracted from the main ligand-into-protein experiment data. $K_d$ values were determined from integrated binding isotherms using the "One set of sites" model in Origin 7.0—based Malvern/MicroCal data analysis software.

## Fluorescence affinity measurements

FP experiments were performed by incubating purified UBE2G2 with 10 nM FITC-labeled G2BR peptide (Peptide 2.0) for 15 minutes at 25°C in 50 mM Tris-HCl pH 7.5, 1 mM DTT, 0.1 mg/ml BSA. Fluorescence anisotropy at 490-nm excitation and 515-nm emission were measured for each titration point in triplicate in Corning 384 well round bottom black polystyrene plates at 25°C with a Spectra Max M5 Microplate Reader (Molecular Devices, San Jose, CA, USA). MST was performed with titration of purified UBE2G2 with FITC-labeled G2BR peptide (Peptide 2.0) in Monolith NT.115 (NanoTemper Technologies, München, Germany). Measurements were repeated at least twice with biological triplicates. Data analysis was carried out using Prism software (GraphPad, San Diego, CA, USA) and fitted by nonlinear least squares assuming 1:1 stoichiometry to estimate the $K_d$.

## Supporting information

**S1 Fig. (A)** gp78 KO cells were transfected with plasmid encoding WT or G2BR mutant gp78 and MYC-tagged INSIG-1, and assessed for INSIG-1 degradation by CHX chase. **(B)** Long exposure of FLAG immunoblot from Fig 1F demonstrating the relatively low expression of the AUP1 ΔATΔCUE mutant. **(C)** AUP1 KO cells were transfected with plasmids encoding FLAG-tagged AUP1 WT or ΔATΔCUE mutant and transcript levels determined following isolation of RNA from cells and qPCR to amplify the FLAG-tagged constructs. Expression of the ΔATΔCUE mutant is presented relative to WT AUP1. Mean and standard deviation are shown. **(D)** HT1080 parental and AUP1 KO cells were allowed to accumulate HMGCR in LPDS media for 24 hours. Basal degradation of HMGCR was assessed by addition of CHX in LPDS media. **(E)** HT1080 AUP1 KO cells were transfected with the indicated plasmids and switched to LPDS media 36 hours post-transfection. After 24 hours, cells were treated with

CHX in LPDS media to assess basal degradation of HMGCR. The data underlying this figure can be found in **S2 and S3 Data**. AUP1, ancient ubiquitous protein 1; CHX, cycloheximide; G2BR, UBE2G2 Binding Region; HMGCR, 3-hydroxy-3-methylglutaryl CoA reductase; KO, knockout; LPDS, lipoprotein-deficient serum; qPCR, quantitative polymerase chain reaction; WT, wild type.
(TIF)

**S2 Fig. (A)** HT1080 or the indicated KO cells were transfected with plasmid encoding $RI_{332}$-MYC and degradation assessed by CHX chase. **(B)** HT1080 UBE2G2 KO cells were transfected with $RI_{332}$-MYC and with either vector or MYC-UBE2G2. $RI_{332}$ stability was assessed as in (A). **(C)** HT1080 and UBE2G2 KO cells were assessed for indicators of an ER stress response by western blot. Total EIF2$\alpha$ and actin serve as internal controls. **(D)** HT1080 parental and UBE2G2 KO cells were assessed for relative transcript levels of ER stress markers by qPCR. For each marker, expression is presented relative to HT1080. Mean and standard deviation are shown ($^*P < 0.05$). **(E)** (Left panel) indicated cells were transfected with plasmid encoding HA-ubiquitin and either empty vector (lanes 1, 4, and 7) or HRD1-MYC-6His (lanes 2, 5, and 8), lysed in urea buffer and pulled down with nickel ($Ni^{2+}$) beads. The supernatant was then subjected to a second pull-down (lanes 3, 6, and 9) with nickel beads to confirm the efficiency of the first pull-down. Eluted samples were immunoblotted for HA-ubiquitin and HRD1-MYC-6His. Inputs of vector transfected (lanes 1, 4, and 7), HRD1-MYC-6His transfected (lanes 2, 5, and 8), and second pull-down (lanes 3, 6, and 9) are shown in the right panel. **(F)** Ubiquitinated proteins were enriched with a mixture of TUBE1 and TUBE2 agarose and treated with buffer or a cocktail of deubiquitinating enzymes. Agarose (UM400) beads served as a control. After extensive washing, proteins were eluted with 2X SDS sample buffer and resolved by SDS-PAGE and immunoblotted for HRD1. The data underlying this figure can be found in **S2 and S4 Data**. CHX, cycloheximide; KO, knockout; qPCR, quantitative polymerase chain reaction.
(TIF)

**S3 Fig. (A)** ITC titration curves from experiment performed with purified UBE2G2 and G2BR$^{gp78}$ (left) or G2BR$^{AUP1}$ (right) peptides. **(B)** FITC-labeled G2BR$^{gp78}$ or G2BR$^{AUP1}$ peptides were incubated with increasing concentrations of purified UBE2G2 at 22˚C, and binding was assessed by MST to determine dissociation constants ($K_d$) between the G2BR and UBE2G2. **(C)** Binding was assessed as in (B) using FP. The data underlying this figure can be found in **S5–S7 Data**. AUP1, ancient ubiquitous protein 1; FP, fluorescence polarization; G2BR, UBE2G2 Binding Region; ITC, isothermal titration calorimetry; MST, microscale thermophoresis.
(TIF)

**S4 Fig. (A)** GST fusions to the RING domain of indicated E3s were incubated with E1, UBE2G2, Cy5-labeled ubiquitin, and either a scrambled G2BR (SCR), G2BR$^{gp78}$, or G2BR$^{AUP1}$ peptide for 1.5 hours at 37˚C. Reactions were resolved by SDS-PAGE and Cy5-ubiquitin was visualized on a phosphorimager. **(B)** Ubiquitination assays described in (A) were carried out with GST-gp78 RING for the indicated times with the specified G2BR peptides. The data underlying this figure can be found in **S2 Data**. AUP1, ancient ubiquitous protein 1; G2BR, UBE2G2 Binding Region.
(TIF)

**S5 Fig. (A)** HT1080 cells were treated with 200 μM oleic acid (OA) overnight prior to fixation and staining to visualize lipid droplets (green), filamentous actin (red), and nuclei (blue) by confocal microscopy. Maximum intensity projection images (without deconvolution) of representative fields are shown. **(B)** HT1080 cells were treated with oleic acid as in (A) and assessed

for degradation of transfected NHK-HA and endogenous UBE2G2. **(C)** To determine expression of GFP-G2BR fusions in transfection experiments, AUP1 KO cells were transfected with plasmid encoding GFP-G2BR$^{gp78}$ and dilutions of lysate were resolved by SDS-PAGE alongside HT1080 lysate. Detection was carried out with affinity-purified antibodies directed against the G2BR of gp78 (Ab2) and compared to levels of endogenous gp78. The level of transfected GFP-G2BR is approximately 40-fold that of endogenous gp78. **(D)** Levels of UBE2G2 were monitored by CHX chase in HEK293 (left panel) and M17 (right panel) parental and AUP1 KO cells. **(E)** HEK293 AUP1 KO cells were transfected with plasmids encoding MYC-UBE2G2 and GFP fusions of either a scrambled G2BR (SCR), G2BR$^{gp78}$, or G2BR$^{AUP1}$, and the turnover of UBE2G2 was assessed. **(F)** Approximately $1 \times 10^4$ cell equivalents of the indicated cell lines were assessed for levels of AUP1 and gp78. Using the 40:1 ratio of the two proteins in HT1080 derived from Fig 7E, approximate relative levels of AUP1 to gp78 were calculated for each cell line. The data underlying this figure can be found in **S2 Data**. AUP1, ancient ubiquitous protein 1; CHX, cycloheximide; G2BR, UBE2G2 Binding Region; KO, knockout; NHK, Null Hong Kong.
(TIF)

**S6 Fig. (A)** HT1080 or AUP1 KO cells were transfected with NHK-HA and either empty vector or FLAG-tagged AUP1 lacking the hairpin transmembrane region (ΔTM), and NHK turnover assessed by CHX chase. **(B)** Relative levels of endogenous (HT1080) or overexpressed UBE2G2 from t = 0 time points in Fig 8C (lanes 1 to 3) and 8H (lanes 4 and 5) in which AUP1 KO cells were transfected with MYC-UBE2G2 or TM-UBE2G2 and GFP fusions of either a scrambled G2BR (SCR) or G2BR$^{AUP1}$. **(C)** HT1080 cells were transfected with plasmid encoding TM-UBE2G2 to which a FLAG tag had been added at the carboxyl terminus to facilitate co-immunoprecipitation. Immunoprecipitates from digitonin lysates were blotted for associated HRD1 and SEL-1L. Inputs are shown on the left. **(D)** UBE2G2 KO cells were transfected with NHK-HA and either empty vector, MYC-UBE2G2 or TM-UBE2G2, and NHK turnover assessed by CHX chase. The data underlying this figure can be found in **S2 Data**. AUP1, ancient ubiquitous protein 1; CHX, cycloheximide; G2BR, UBE2G2 Binding Region; KO, knockout; NHK, Null Hong Kong.
(TIF)

**S1 Data. Raw images from protein gel electrophoresis in Figs 1–8.**
(PDF)

**S2 Data. Raw images from protein gel electrophoresis in S1–S6 Figs.**
(PDF)

**S3 Data. Raw and calculated qPCR data for S1C Fig comparing transcript levels of the AUP1 WT and ΔATΔCUE mutant from plasmid transfection.** AUP1, ancient ubiquitous protein 1; qPCR, quantitative polymerase chain reaction; WT, wild type.
(XLSX)

**S4 Data. Raw and calculated qPCR data for S2D Fig comparing expression of HRD1 and several UPR markers in HT1080 parental and UBE2G2 KO cells.** KO, knockout; qPCR, quantitative polymerase chain reaction.
(XLSX)

**S5 Data. Raw ITC data for S3A Fig showing molar ratios as a function of time (seconds).** ITC, isothermal titration calorimetry.
(XLSX)

**S6 Data. Raw MST data for S3B Fig.** MST, microscale thermophoresis.
(XLSX)

**S7 Data. FP data for S3C Fig.** Raw data showing changes in FP with E2 titration. FP, fluorescence polarization.
(XLSX)

**S8 Data. Raw and calculated qPCR data for Fig 7B comparing UBE2G2 expression in HT1080 parental and AUP1 KO cells.** AUP1, ancient ubiquitous protein 1; KO, knockout; qPCR, quantitative polymerase chain reaction.
(XLSX)

**S1 Table. Data corresponding to NMR experiments in Fig 5.** NMR, nuclear magnetic resonance.
(PDF)

## Acknowledgments

We thank Jadranka Loncarek for helpful discussion and assistance with microscopy and Stanley Lipkowitz and Meredith Metzger for critical reading of the manuscript. We also thank members of the NCI-Frederick Flow Cytometry Core for cell sorting.

## Author Contributions

**Conceptualization:** Yien Che Tsai, Allan M. Weissman.

**Data curation:** Christopher E. Smith, Yien Che Tsai, Yu-He Liang, R. Andrew Byrd, Xinhua Ji.

**Formal analysis:** Christopher E. Smith, Yien Che Tsai, Yu-He Liang, Domarin Khago, Sergey G. Tarasov, R. Andrew Byrd, Xinhua Ji.

**Funding acquisition:** R. Andrew Byrd, Xinhua Ji, Allan M. Weissman.

**Investigation:** Christopher E. Smith, Yien Che Tsai, Yu-He Liang, Domarin Khago, Jennifer Mariano, Jess Li, Sergey G. Tarasov, Emma Gergel, Borong Tsai, Matthew Villaneuva.

**Methodology:** Yien Che Tsai, Yu-He Liang, Sergey G. Tarasov, Valentin Magidson, Raj Chari, R. Andrew Byrd, Xinhua Ji.

**Project administration:** Allan M. Weissman.

**Resources:** Jess Li, Michelle E. Clapp, Raj Chari, R. Andrew Byrd, Xinhua Ji.

**Supervision:** Yien Che Tsai, Valentin Magidson, R. Andrew Byrd, Xinhua Ji, Allan M. Weissman.

**Visualization:** Christopher E. Smith, Yien Che Tsai, Yu-He Liang, Domarin Khago, Jennifer Mariano, Valentin Magidson, R. Andrew Byrd, Xinhua Ji.

**Writing – original draft:** Christopher E. Smith, Yien Che Tsai, Yu-He Liang, Jennifer Mariano, R. Andrew Byrd, Xinhua Ji, Allan M. Weissman.

**Writing – review & editing:** Christopher E. Smith, Yien Che Tsai, Allan M. Weissman.

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
