## [Editor Report · Decision Letter 0]

8 Feb 2021

Dear Dr Weissman, 

Thank you for submitting your manuscript entitled "A structurally conserved binding site for UBE2G2 in ancient ubiquitous protein 1 (AUP1) is essential for endoplasmic reticulum-associated degradation (ERAD)" for consideration as a Research Article by PLOS Biology.

Your manuscript has now been evaluated by the PLOS Biology editorial staff as well as by an academic editor with relevant expertise and I am writing to let you know that we would like to send your submission out for external peer review.

Please re-submit your manuscript within two working days, i.e. by Feb 10 2021 11:59PM.

Kind regards,

Richard

Richard Hodge, PhD

Associate Editor

PLOS Biology

---

## [Decision Letter · Decision Letter 1]

25 Mar 2021

Dear Dr Weissman,

Thank you very much for submitting your manuscript "A structurally conserved binding site for UBE2G2 in ancient ubiquitous protein 1 (AUP1) is essential for endoplasmic reticulum-associated degradation (ERAD)" for consideration as a Research Article at PLOS Biology. Please accept my apologises for the delays you have experienced in the processing of your manuscript. Your manuscript has been evaluated by the PLOS Biology editors, an Academic Editor with relevant expertise, and by several independent reviewers.

The reviews are attached below. You will see that the reviewers find your conclusions interesting and the data well executed, but they also ask for some overlapping experiments that would strengthen the conclusions. Specifically, Reviewer #1 and #2 suggest additional control experiments investigating Hrd1-dependent substrates. In addition, Reviewer #1 and #3 ask that the initial observations are repeated in additional cell lines and whether AUP1 binding can be titrated away from UBE2G2.

In light of the reviews, we will not be able to accept the current version of the manuscript, but we would be delighted to consider a revision that takes into account the reviewers' comments. We cannot make any decision about publication until we have seen the revised manuscript and your response to the reviewers' comments. Your revised manuscript is also likely to be sent for further evaluation by the reviewers.

We expect to receive your revised manuscript within 2 months. 

**IMPORTANT - SUBMITTING YOUR REVISION**

*Re-submission Checklist*

*Published Peer Review*

*PLOS Data Policy*

*Blot and Gel Data Policy*

Sincerely,

Richard

Richard Hodge, PhD

Associate Editor, PLOS Biology

rhodge@plos.org

REVIEWS:

Reviewer's Responses to Questions

PLOS authors have the option to publish the peer review history of their article (what does this mean?). If published, this will include your full peer review and any attached files.

Reviewer #1: No

Reviewer #2: Yes: John C Christianson

Reviewer #3: No

Reviewer #1: The manuscript submitted to PLoS Biology by Weissman and colleagues reports on a long awaited analysis of the mechanism underlying the role of AUP1 in ERAD, and more specifically on its molecular activation of partner E3 ubiquitin ligases via an embedded G2BR. 

A related domain, U7BR was previously analyzed by this group as a "back-side" binding motif in yeast Cue1 for a partner ubiquitin conjugating enzyme, Ubc7. In addition, in the mammalian E3, gp78, a similar motif, G2BR, bound and activated an analogous ubiquitin conjugating enzyme, UBE2G2. Interestingly, a Cue1 analog in higher cells, AUP1, has been implicated in not only the ubiquitination of some ERAD substrates (although its substrate specificity is somewhat controversial), but in lipid droplet homeostasis by virtue of an acyltransferase activity. The CUE domain may interact with both ubiquitinated substrates as well as lipids. Similar to what was observed for Cue1 and gp78, AUP1 also interacts with UBEG2 via an embedded G2BR, but the role of this motif and the acyltransferase moiety in ERAD, let alone other AUP-associated functions, has not been investigated.

To this end, Smith et al. first find that AUP1 is required for the degradation of three diverse ERAD substrates, an effect that depended on G2BR but not on the acyltransferase or CUE1 domains in AUP1. As expected, the ubiquitination of one substrate (but not HRD1) also required the domain. The authors next investigated the specificity of G2BR from AUP1 and the homologous G2BR in gp78. The G2BR from AU1P bound in vitro translated UBEG2 but neither UBEG1 (which binds the gp78 G2BR) nor several "promiscuous" E2s. To further investigate the mechanism underlying UBE2G2-G2BR specificity and function, the structure of this complex was solved by X-ray crystallography. As anticipated, the structures of the G2BRs from AUP1 and gp78 were similar and a high affinity (~3 nM) interaction, mediated by select amino acids, with UBE2G2 was measured. NMR spectroscopy indicated that the G2BR from AUP1, like that from gp78, alters the conformation of the E2. Based on the measured affinity of the G2BR and comparisons to RING domain association with the E2, the resulting conformational change brought about by the AUP1 G2BR increased the affinity of the E2 for the gp78 RING domain by ~10-fold, an effect that was reflected in enhanced auto-ubiquitinated of two examined (HRD1 and TRC8) ligases. Next, the authors asked whether—as might be anticipated by backside, allosteric activation of the E2 and ligase activity—the G2BR-dependent interaction of UBEG2 with AUP1 plays a role in ERAD. Mutation of 5 polar/charged amino acids in the interaction site abrogated in vitro binding, and similarly failed to support the ERAD of one of the AUP1-dependent substrates (NHK) studied earlier. Mutation of two key hydrophobic residues to Asp had similar effects. A single mutant provided some hint of an intermediate effect on binding (but see below). Based on results from the analogous factors in yeast, Smith et al. then asked if AUP1 protects its partner E2 from degradation. Stability was mediated by either the AUP1 or—more surprisingly—the gp78 G2BR. Nevertheless, UBE2G2 remained stable in gp78 knockout cells, which might arise from its significantly higher levels, at least in HT1080 cells (also see below). Finally, the link between the requirement for AUP1 in HRD1-dependent degradation was investigated. After ruling a potential artefact, the authors focused on the potential role of AUP1 is not only in the stabilization of UBE2G2 but in recruiting the E2 to the ER membrane. A striking result was obtained when a synthetic ER-tethered form of UBE2G2 was found to rescue AUP1 via G2BR association. These results indicate that yet another role of AUP1 is to harness the E2 to the membrane.

Overall, this is a well written, thorough, and logically presented story that provides new insights into the specific interactions amongst ERAD-associated factors that are required for substrate ubiquitination. The results of this study more specifically support three independent functions of the G2BR in AUP1. The data are exceptionally clean and controlled and utilized diverse techniques. While some of the results were expected based on prior research on the analogous proteins in yeast, there are several striking differences between AUP1 and Cue1 that warrant attention. Thus, the study will be of interest to readers in several fields who use various models.

Comments:

1. In the author's hands, is mutant rhodopsin also a HRD1 substrate (viz. Fig. 1)? In fact, the first sentence of the Discussion indicates that the requirements for the degradation of RI332 were explored in this study, which is not true. In addition to the HRD1 control, the conclusions of this study would be strengthened if select experiments described later in the text utilized this second substrate.

2. The data in Figure S4 suggest not only increased auto-ubiquitination by also increased processivity. This effect could be better resolved by showing a time-course.

3. There is a hint from the data in Figure 5D that the A397D mutant, which significantly reduces but does not block binding, may allow for limited ERAD of NHK. Since the blots are overexposed and only a single experiment is shown, have the authors thought about examining whether binding and the associated support of ERAD is a titratable phenomenon?

4. A better test of the model that the levels of gp78 relative to AUP1 dictate which G2BR-containing protein stabilizes UBE2G2 (Figure 6) is to examine the levels of the two proteins in other cell lines. More likely than not, perhaps the ratio of gp78 to AUP1 differs? Indeed, components involved in cellular quality control vary widely between cells.

Minor comments

5. Since the ubiquitination pattern of HRD1 is clearly delineated as starting at ~75 kD (Fig. 2F), the authors should note the molecular mass of this E3.

6. In the relevant section of the text, the reader should be referred to the affinities calculated using FP, which are presented in Figure 3D.

7. The x-axis in Figure 4E lacks a label.

8. Without explanation, it appears that U2OS cells were used for the experiment in Figure 7D. What was the rationale for this switch? The data linking the results in Figure S5 (in HT1080 cells) to this figure would best to compared with identical cell types. 

9. Perhaps Figures S5C-D, which provide key verification of the chimera, could accompany the results within Figure 7.

Reviewer #2 (John C Christianson): MS #: E17-08-0514

TITLE: A structurally conserved binding site for UBE2G2 in ancient ubiquitous protein 1 (AUP1) is essential for endoplasmic reticulum-associated degradation (ERAD)

AUTHORS: Smith et al. 

OVERVIEW

In this manuscript, the authors describe a multifaceted role for the G2BR binding site of AUP1 in ER-associated degradation (ERAD). They build on a variety of data from their lab and other reports to characterise this domain in detail and provide new insights into the role that this interaction might be playing in ERAD. Confirming screening data reported previously, the authors confirm essential roles for both AUP1 and Ube2g2 in turnover of ERAD-L (NHK & RI332) and ERAD-C (GFPu) substrates using CRISPR KOs. The authors demonstrate ERAD of model substrates can be rescued by full length AUP1 and almost all truncations/deletions/mutants that do not disrupt the G2BR or disconnect it from the ER membrane. Ubiquitination of substrates but not the E3 Hrd1 is attenuated in KOs of both AUP1 and Ube2g2, leading the authors to differentiate its role from that of another E2, Ube2j1. The authors use recombinant proteins and in vitro assays to show specificity of Ube2g2 for AUP1 and its G2BR. ∆Ube2g2 but not ∆AUP1 cells exhibit defects in degradation of Insig1, a gp78-dependent membrane protein. The authors also provide NMR data which supports a finding where the G2BR domain in AUP1 behaves comparably to that found in gp78, going on to identify residues supporting key electrostatic and hydrophobic interactions between AUP1 and Ube2g2. AUP1 variants containing mutations to key residues in the G2BR responsible for Ube2g2 binding as determined by NMR, lost AUP1 binding and were not able to rescue NHK degradation. Like the G2BR-gp78, the authors find, perhaps not unexpectedly, that G2BR-AUP1 also activates Ube2g2. Interestingly, they do find that interaction with AUP1-G2BR rather than gp78-G2BR has prominent role in protecting Ube2g2 by maintaining its stability and keeping it associated with the ER membrane. The authors attribute this to a ~40-fold higher abundance of AUP1. The authors provide evidence that Ube2g2 minimally requires the high-affinity G2BR binding and ER recruitment to functionally restore ERAD, with the acyltransferase and upstream ubiquitin-binding CUE domain, entirely dispensable. These data lead the authors to conclude a multifaceted role for AUP1 and its G2BR - to activate Ube2g2, to keep it from self-targeting for degradation, and to recruit it to the ER membrane, collectively serving to ubiquitinate ERAD substrates. 

Overall the manuscript is suitable in detail and scope for the journal. While not presenting entirely novel concepts, what the authors have done is to put together a thorough collection of experiments that clarifies a lot of previous data to yield a clearer picture of the role that AUP1, its G2BR and Ube2g2 play in ERAD. There are several intriguing implications from these data, which will help to guide the field to a better understanding of the ERAD mechanism, while also providing general appeal to an audience keen to understand complex, regulated ubiquitination mechanisms. The manuscript is clearly written and appropriately referenced. In this reviewer's view, there are just a few issues that could require attention.

SELECTED ISSUES 

Issue 1. The authors provide data demonstrating the ERAD-L substrates NHK and RI332 require both AUP1 and Ube2g2 and Hrd1, while Insig-1, an ERAD-M substrate only requires Ube2g2 and gp78. The clear omission here is the absence of a Hrd1-dependent ERAD-M substrate to directly confirm the AUP1-Ube2g2 roles in ERAD. The breadth of the conclusions would be greater if this could be shown, perhaps simply by probing for endogenous CD147 and monitoring its ER-resident form (see Tyler et al. JBC. 2011 or Schulz et al. JCS 2017) or MHC Class I with B2m silencing (see Burr et al. PNAS 2011).

Issue 2. In line 335, the author ascertain that there is nearly 40-fold more AUP1 relative to gp78 in HT1080 cells. This claim prompts several questions. First, is the amount of GFP-G2BR expressed to protect Ube2g2 in ∆AUP1 cells (Figure 6D) near to the ~40x amount of endogenous gp78? Secondly, Figure S5C shows increased cytosolic partitioning of Ube2g2 in ∆AUP1 cells compared to ∆gp78 or WT, but this amount does not appear to be 40x different as might be expected, and more cytosolic. Does this imply that Ube2g2 levels might be rate limiting in normal cells, with competition between gp78 and AUP1 for binding? Perhaps the authors could comment on this. 

Issue 3. In Figure 7, the authors provide data supporting the idea that mere tethering of the Ube2g2 to the ER membrane (TM-UBE2G2) via an unrelated TMD along with a soluble AUP1-G2BR is sufficient to restore Hrd1-dependent degradation of NHK. Are the authors implying that interaction/s of AUP1 with the Hrd1 complex are dispensable for this process? This would be slightly surprising to this review so perhaps the authors might expand further on the implications here. 

SPECIFIC COMMENTS/QUESTIONS

Comment 1. For the ubiquitination assays, the authors use concentrations of MG132 that are unexpectedly high (e.g. 40µM) and well into the range of non-selective effects. Is there a reason why better ERAD inhibitors, such as CB-5083 (@ 10nM) or bortezomib (@25nM), were not used? 

Comment 2. In Figure 2E, ER stress induction is proposed as the reason for increased Hrd1 levels in Ube2g2 KO cells, but this is not directly demonstrated. Would it be possible to perform a simple XBP-1 splicing assay to confirm this or to also probe for another ER stress regulated gene such as Herp or Bip? This would help to solidify this point. 

Comment 3. Previous studies have reported AUP1 to be mono- and di-ubiquitinated (Klemm et al. JBC 2011), yet this form does not appear to be present in any of the western blots presented. Could the authors address why this discrepancy might exist?

Comment 4. The authors present Hrd1 ubiquitination data for both transfected and endogenous forms, which is notable in that smears do not appear to change under any conditions. This reviewer would feel more reassured with these data if smears could be at least shown to be sensitive to recombinant DUB treatment or lost from cells where Ube2j1/Ubc6e is knocked down or out, as has been suggested by other data in the manuscript. 

Reviewer #3: The article by Smith, C. E. et al., describes an important role for ancient ubiquitous protein 1 (AUP1) and its interaction with the E2 conjugating enzyme, through its C-terminal G2BR domain, in ER-associated degradation (ERAD). Using a fibrosarcoma cell line (HT-10180), the authors initially describe the requirement for AUP1 in HRD1-mediated degradation of the Null Hong Kong (NHK) variant of alpha1-antitrypsin. They also provide evidence, through deletion mutations, that AUP1's most significant contribution to the degradation of ERAD substrates is from its UBE2G2 binding region (G2BR) and that the acyltransferase and Cue1 domains are dispensable. Membrane and luminal ERAD of INSIG1 and NHK (respectively) is shown to require functional UBE2G2 for ubiquitination (NHK and HRD1) and turnover (NHK and INSIG1). 

The authors elegantly define the interaction between UBE2G2 and AUP1 using crystallography, mutation of residues and biochemistry. While comparing the structure and interaction of UBE2G2 and AUP1's G2BR domain they also highlight how similar the interaction and kinetics are to that of gp78's, G2BR interaction with UBE2G2. The authors report that many of the amino acid residues required for the interaction of AUP1 and UBE2G2 are hydrophobic and show that three specific residues L386, Y394 and A397, when mutated to aspartic acid, prevent the interaction between UBE2G2 and AUP1.

The interaction of AUP1 with UBE2G2 appears to stabilise levels of UBE2G2 in the HT-1080 cell line and only deletion of AUP1 and not gp78 de-stabilises UBE2G2. This result is not due to significant changes in the mRNA levels but does require UBE2G2 activity - catalytically dead C89S UBE2G2 does not exhibit the same phenotype. Confusingly, addition of GFP-tagged gp78 or AUP1 G2BRs resulted in the stabilisation of UBE2G2. The authors reconcile this data by showing that AUP1's role in UBE2G2 stability is a result of higher levels of expression compared to gp78. Finally, they show that AUP1 is dispensable if a transmembrane localised UBE2G2 is expressed alongside GFP-tagged AUP1 G2BR. Taken together this suggests the minimum requirements for NHK turnover are an ER membrane bound protein that can interact with or is itself UBE2G2 and a G2BR. If either is missing UBE2G2 cannot be utilised in ERAD. 

This is a well-planned, executed and expansive paper on the role of G2BRs in gp78 and AUP1 and sheds light on a role of AUP1 in ERAD that has not been previously demonstrated. There are a few issues that arise that could be address to make this paper much stronger and make it more accessible to a larger readership.

Major points:

The phenotypes they show are strong but it would be nice to see the initial observations repeated in other cell lines.

Although, it is possible that the stability of UBE2G2 is due to the level of AUP1 in these cells, it is not clear whether the G2BR of AUP1 is always available for UBE2G2 binding and whether AUP1 acts as a sink for UBE2G2 when associated with lipid droplets. As the authors state, AUP1 has multiple roles in cells, and therefore a clear experiment that is needed here is immunoprecipitation of endogenous UBE2G2 followed by a blot for AUP1 and gp78 to show that they bind different levels of UBE2G2 under the conditions used by the authors in these experiments. They can then compare the 'pulldown levels' of AUP1 and gp78 to the amounts of protein in the reference binding domains in (Figure 6).

It would also help their assertion that 'more AUP1 equals more UBE2G2 due to G2BR binding', if they could show that AUP1 binding can be titrated away from UBE2G2 when cells are fed increasing amounts of Oleic acid, or whether this interaction is unaffected as AUP1 is re-localised to lipid droplets. Alternatively, NHK degradation may be reduced under increasing Oleic acid levels but UBE2G2 binding is not affected. This would provide further evidence that AUP1 is indeed a protective sink for UBE2G2 under multiple conditions and that protein levels are a significant reason for why AUP1 is important for preventing UBE2G2 turnover.

Although U2OS cells provide nice results when looking at ER and proteins that localise to the ER, the authors have not shown that their mechanism holds in this cell line and do not use them throughout the paper other than to show localisation of their Myc-tagged TM-UBE2G2. Therefore, it is important to either confirm a similar phenotype for these cells or show the distribution of this protein (alongside an ER resident) in the HT-1080 cells. 

Minor points:

It is unclear why the GFP/Luciferase blots are included as transfection controls. Unless the GFP is upstream or downstream of the protein of interest or connected via a viral P2A sequence it will not directly report on how well a transfection has occurred for the protein of interest. I cannot see a reason for including this control in most blots and in several it probably suggests there is clear disparity between 2 cell types even though the protein(s) of interest are expressed at quite similar levels.

---

## [Decision Letter · Decision Letter 2]

6 Oct 2021

Dear Dr Weissman,

Thank you for submitting your revised Research Article entitled "A structurally conserved binding site for UBE2G2 in ancient ubiquitous protein 1 (AUP1) is essential for endoplasmic-reticulum associated degradation (ERAD)" for publication in PLOS Biology. I have now obtained advice from the original reviewers and have discussed their comments with the Academic Editor. 

The reviewers note that the additional data provided in the revision has further strengthened the manuscript and they appreciate the time and effort that was spent in addressing their concerns. Based on these reviews, we will probably accept this manuscript for publication, provided you satisfactorily address our remaining data and policy-related requests which are provided below:

(A) We would like to suggest the following change to the title of the manuscript, to make it clearer for our broad readership:

‘A structurally conserved binding site in AUP1 enables binding to the E2 enzyme UBE2G2 and is essential for ER-associated degradation’

(B) You may be aware of the PLOS Data Policy, which requires that all data be made available without restriction: http://journals.plos.org/plosbiology/s/data-availability. For more information, please also see this editorial: http://dx.doi.org/10.1371/journal.pbio.1001797

- Supplementary files (e.g., excel). Please ensure that all data files are uploaded as 'Supporting Information' and are invariably referred to (in the manuscript, figure legends, and the Description field when uploading your files) using the following format verbatim: S1 Data, S2 Data, etc. Multiple panels of a single or even several figures can be included as multiple sheets in one excel file that is saved using exactly the following convention: S1_Data.xlsx (using an underscore).

- Deposition in a publicly available repository. Please also provide the accession code or a reviewer link so that we may view your data before publication.

Regardless of the method selected, please ensure that you provide the individual numerical values that underlie the summary data displayed in the following Figures, as they are essential for readers to assess your analysis and to reproduce it:

Figure 5A-E, 7B, S1C, S2D, S3A-C

(C) Please also ensure that each of the relevant figure legends in your manuscript include information on *WHERE THE UNDERLYING DATA CAN BE FOUND*, and ensure your supplemental data file/s has a legend

(D) Please ensure that your Data Statement in the submission system accurately describes where your data can be found and is in final format, as it will be published as written there. At this time, we ask that you please make sure the data deposited at the PDB (7LEW) is made publicly available, as it is currently on hold. 

(E) Thank you for sending the original raw gels, but please note that we require the raw uncropped gels for all instances and in some cases, you cannot see the full gel. We would be grateful if you could modify the raw image document to provide the full uncropped images in these specific instances, such as for Figure S2E (MYC-HRD1) and Figure S1E (FLAG-AUP1).

-------------------

We expect to receive your revised manuscript within two weeks. 

*Published Peer Review History*

*Early Version*

Sincerely,

Richard

Richard Hodge, PhD

Associate Editor, PLOS Biology

rhodge@plos.org

Reviewer remarks:

Reviewer #1:

The authors have done an excellent job at addressing all comments on the prior submission.

Reviewer #2: John Christianson - note that this reviewer has signed his review

The authors have satisfactorily addressed my minor comments and suggestions in their revision. This reviewer appreciates the extra time and effort that went into improving the manuscript and hopes that it has broadened the impact of its findings. 

Reviewer #3:

This was already a strong manuscript and the further data has strengthened it further - I would therefore recommend publication

---

## [Editor Report · Decision Letter 3]

29 Oct 2021

Dear Dr Weissman,

Thank you for submitting your revised Research Article entitled " A structurally conserved binding site in AUP1 enables binding to the E2 enzyme UBE2G2 and is essential for ER-associated degradation" for publication in PLOS Biology and for addressing our editorial requests. Please accept my apologies, but we ask that you please provide some additional information before your manuscript can be accepted for publication.

To submit your revision, please go to https://www.editorialmanager.com/pbiology/ and log in as an Author.

Click the link labelled 'Submissions Needing Revision' to find your submission record. Your revised submission must include the following:

- a cover letter that should detail your responses to any editorial requests

DATA POLICY

• We note that the data deposited at the PDB (7LEW) is currently still on hold. We ask that you please make this data publicly available before we accept your manuscript. 

• Thank you for providing the underlying data for the Figures in the Supplementary data files. However, we note that the raw data for the NMR data presented in Figures 5A-C has not been provided. Since this may represent a large amount of data, we ask that you please deposit this data in a repository. We recommend the Biological Magnetic Resonance Data Bank ((https://bmrb.io/). Please include the accession number and, as before, ensure that the data is made publicly available at this stage. 

• In your Supplementary data files (S3-S8), please label the tabs of the spreadsheet or provide headings so it is made clear what figure corresponds to each dataset. 

• We note that your Data Availability statement currently states:

‘Crystallography coordinates have been deposited with the RCSB protein data bank (PDB entry 7LEW) and will be available upon acceptance’

Please update this statement in the submission system to ensure that it accurately describes where your data can be found and is in final format, as it will be published as written there. This includes referencing the raw data in the Supplementary Files provided in your submission and the data deposited in the BMRB and PDB (with accession numbers).

Please do not hesitate to e-mail me directly at rhodge@plos.org if you have any questions or concerns. As soon as the points above have been addressed, we will be able to accept your manuscript for publication.

Sincerely,

Richard

Richard Hodge, PhD

Associate Editor, PLOS Biology

rhodge@plos.org

PLOS

---

## [Editor Report · Decision Letter 4]

5 Nov 2021

Dear Dr Weissman,

On behalf of my colleagues and the Academic Editor, Raquel Lieberman, I am pleased to say that we can in principle accept your Research Article "A structurally conserved binding site in AUP1 enables binding to the E2 enzyme UBE2G2 and is essential for ER-associated degradation" for publication in PLOS Biology, provided you address any remaining formatting and reporting issues. These will be detailed in an email that will follow this letter and that you will usually receive within 2-3 business days, during which time no action is required from you. Please note that we will not be able to formally accept your manuscript and schedule it for publication until you have any requested changes.

PRESS

Sincerely, 

Ines

--

Ines Alvarez-Garcia, PhD

Senior Editor

PLOS Biology

on behalf of

Richard Hodge, PhD

Senior Editor 

PLOS Biology

rhodge@plos.org